# Unsupervised Learning for Combinatorial Optimization with Principled Objective Relaxation

**Haoyu Wang**
Purdue University
wang5272@purdue.edu

**Nan Wu**
UCSB
nanwu@ucsb.edu

**Hang Yang**
Georgia Tech.
hyang628@gatech.edu

**Cong Hao**
Georgia Tech.
callie.hao@gatech.edu

**Pan Li**
Purdue University
panli@purdue.edu

## Abstract

Using machine learning to solve combinatorial optimization (CO) problems is challenging, especially when the data is unlabeled. This work proposes an unsupervised learning framework for CO problems. Our framework follows a standard relaxation-plus-rounding approach and adopts neural networks to parameterize the relaxed solutions so that simple back-propagation can train the model end-to-end. Our key contribution is the observation that if the relaxed objective satisfies entry-wise concavity, a low optimization loss guarantees the quality of the final integral solutions. This observation significantly broadens the applicability of the previous framework inspired by Erdős' probabilistic method [1]. In particular, this observation can guide the design of objective models in applications where the objectives are not given explicitly while requiring being modeled in prior. We evaluate our framework by solving a synthetic graph optimization problem, and two real-world applications including resource allocation in circuit design and approximate computing. Our framework[1] largely outperforms the baselines based on naïve relaxation, reinforcement learning, and Gumbel-softmax tricks.

## 1 Introduction

Combinatorial optimization (CO) with the goal of finding the optimal solution from a discrete space is a fundamental problem in many scientific and engineering applications [2–4]. Most CO problems are NP-complete. Traditional methods efficient in practice often either depend on heuristics or produce approximation solutions. Designing these approaches requires considerable insights into the problem. Recently, machine learning has paved a new way to develop CO algorithms, which asks to use neural networks (NNs) to extract heuristics from the data [5–7]. Several learning for CO (LCO) approaches have therefore been developed, providing solvers for the problems including SAT [8–10], mixed integer linear programming [11–13], vertex covering [14, 15] and routing problems [16–23].

Another promising way to use machine learning techniques is to learn proxies of the CO objectives whose evaluation could be expensive and time-consuming [24–27]. For example, to optimize hardware/system design, evaluating the objectives such as computation latency, power efficiency [28], and resource consumption [29–31] requires running complex simulators. Also, in molecule design, evaluating the objective properties such as protein fluorescence or DNA binding affinity needs either costly simulations or living experiments [32–34]. In these cases, proxies of the objectives can be learned first to reduce the evaluation cost [31], and then optimize these proxies to solve the design problems. Later, we name the CO problem with proxy objectives as Proxy-based CO (PCO) problems.

---

[1]Our code and the datasets are available at: `https://github.com/Graph-COM/CO_ProxyDesign`

36th Conference on Neural Information Processing Systems (NeurIPS 2022).

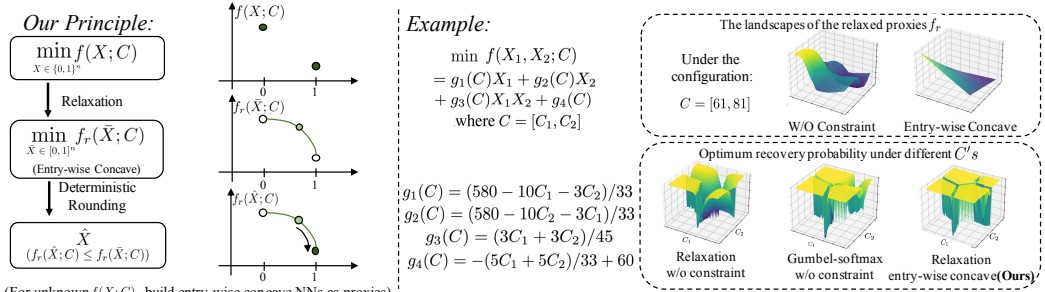

Figure 1: The entry-wise concave relaxation-and-rounding principle (for the case without constraints as an illustration) and an example. Consider an optimization objective $f(X; C)$, $X \in \{0, 1\}^n$. Here, $C$ is the problem configuration such as an attributed graph in a graph optimization problem. We relax the objective to an *entry-wise concave* $f_r(\bar{X}; C)$, $\bar{X} \in [0, 1]^n$. The soft solution by minimizing $f_r(\bar{X}; C)$ will be rounded to a discrete solution (in Def. 1) with performance guarantee. When $f(X; C)$ is not explicitly given, we will learn its relaxed proxies $f_r$ by a NN. This corresponds to a PCO problem. In the toy example on the right, we first learn different relaxed proxies $f_r$'s with or without the entry-wise concave constraint. We compare their landscapes in the top-right figure. We further optimize both proxies and round the obtained soft solutions to integral solutions. The bottom-right figure shows the optimum-recovery probabilities of different methods under different $C$'s.

Moreover, we claim that learning for PCO is often of greater significance than learning for traditional CO problems because commercial CO solvers such as Gurobi cannot applied in PCO due to the missing closed-form objectives and the lack of heuristics for such proxy objectives. Generic solvers such as simulated annealing [35] may be applied while they are extremely slow.

In this work, we propose an unsupervised LCO framework. Our findings are applied to general CO problems while exhibiting extraordinary promise for PCO problems. Unsupervised LCO has recently attracted great attentions [1, 9, 20, 36, 37] due to its several great advantages. Traditional supervised learning often gets criticized for the dependence on huge amounts of labeled data [38]. Reinforcement learning (RL) on the other hand suffers from notoriously unstable training [39]. In contrast, unsupervised LCO is superior in its faster training, good generalization, and strong capability of dealing with large-scale problems [1]. Moreover, unsupervised learning has never been systematically investigated for PCO problems. Previous works for PCO problems, e.g., hardware design [30, 31], were all based on RL. A systematic unsupervised learning framework for PCO problems is still under development.

Our framework follows a relaxation-plus-rounding approach. We optimize a carefully-designed continuous relaxation of the cost model (penalized with constraints if any) and obtain a soft solution. Then, we decode the soft assignment to have the final discrete solution. This follows a common approach to design an approximation algorithm [40, 41]. However, the soft assignment here is given by a NN model optimized based on the historical (unlabeled) data via gradient descent. Note that learning from historical data is expected to facilitate the model understanding the data distribution, which helps with extracting heuristics, avoiding local minima and achieving fast inference. An illustration of our principle with a toy-example is shown in Fig. 1. We also provide the pipeline to empirically evaluate our relaxation-plus-rounding principle in Fig. 2.

Our method shares a similar spirit with [1] while making the following significant contributions. We abandon the probabilistic guarantee in [1], because it is hard to use for general CO objectives, especially those based on proxies. Instead, we design a deterministic objective relaxation principle that gives performance guarantee. We prove that if the objective relaxation is entry-wise concave w.r.t. the binary optimization variables, a low-cost soft solution plus deterministic sequential decoding guarantees generating a valid and low-cost integral solution. This principle significantly broadens the applicability of this unsupervised learning framework. In particular, it guides the design of model architectures to learn the objectives in PCO problems. We also justify the wide applicability of the entry-wise concave principle in both theory and practice.

We evaluate our framework over three PCO applications including feature-based edge covering & node matching problems, and two real-world applications, including imprecise functional unit assignment

in approximate computing (AxC) [42–46] and resource allocation in circuit design [24, 30]. In all three applications, our framework achieves a significant performance boost compared to previous RL-based approaches and relaxed gradient approaches based on the Gumbel-softmax trick [47–49]. The datasets for these applications are also provided in our github repository.

## 1.1 Further Discussion on Related Works

Most previous LCO approaches are based on RL [13, 14, 16–18, 22, 50–52] or supervised learning [11, 12, 38], as these two frameworks do not hold any special requirements on the formulation of CO problems. However, they often suffer from the issues of training instability and subpar generalization. Previous works on unsupervised learning for CO have studied satisfaction problems [9, 36, 53]. Applying these approaches to general CO problems requires problem reductions. Others have considered max-cut [37] and TSP problems [20], while these works depend on carefully selected problem-specific objectives. The work most relevant to ours is [1] and we give detailed comparison in Sec. 3. Note that all previous works on unsupervised learning for CO do not apply to PCO as they need an explicit objective to manipulate. For PCO problems, previous studies focus on how to learn more generalizable proxies of the objectives, such as via Bayesian learning [54, 55] and adverserial training [56, 57]. Once proxies are learned, direct objective relaxation plus gradient descent [56], or RL [30, 31], or general MIP solvers with reformulations of the objectives [58, 59] have been often adopted. Studying generalization of proxies is out of the scope of this work while we conjecture that our suggested entry-wise concave proxies seem smoother than those without constraints (See Fig. 1) and thus have the potential to achieve better generalization.

## 2 Preliminaries and Problem Formulation

In this section, we define several useful concepts and notations.

**Combinatorial Optimization (CO).** Let $C \in \mathcal{C}$ denote a data-based configuration such as a graph with weighted edges. Let $\Omega$ be a finite set of all feasible combinatorial objects and each object has a binary vector embedding $X = (X_i)_{1 \leq i \leq n} \in \{0, 1\}^n$. For example, in the node matching problem, each entry of $X$ corresponds to an edge to denote whether this edge is selected or not. Note that such binary embeddings are applicable even when the choice is not naturally binary: Choosing at most one element from a tuple $(1, 2, 3)$ can be represented as a 3-dim binary vector $(X_1, X_2, X_3)$ with the constraint $X_1 + X_2 + X_3 \leq 1$. W.l.o.g, we assume an algebraic form of the feasible set $\Omega \triangleq \{X \in \{0, 1\}^n : g(X; C) < 1\}$ where $g(X; C) \geq 0$ for all $X \in \{0, 1\}^n$ [2]. For notational simplicity, we only consider one inequality constraint while our later discussion in Sec. 3 and our case studies in Sec. 5 may contain multiple inequalities. Given a configuration $C$ and a constraint $\Omega$, a combinatorial optimization (CO) is to minimize a cost $f(\cdot; C)$ by solving

$$\min_{X \in \{0,1\}^n} f(X; C), \quad \text{s.t.} \quad g(X; C) < 1. \tag{1}$$

**Proxy-based CO (PCO).** In the many applications, the cost or the constraint may not be cheaply evaluated. Some proxies of the cost $f$ or the constraint $g$ often need to be learned from the historical data. With some abuse of notations, we interchangably use $f$ ($g$, resp.) to denote the objective (the constraint, resp.) and its proxy.

**Learning for CO/PCO (LCO).** A LCO problem is to learn an algorithm $\mathcal{A}_\theta(\cdot) : \mathcal{C} \to \{0, 1\}^n$, say a neural network (NN) parameterized by $\theta$ to solve CO or PCO problems. Given a configuration $C \in \mathcal{C}$, we expect $\mathcal{A}_\theta$ to (a) generate a valid solution $\hat{X} = \mathcal{A}_\theta(C) \in \Omega$ and (b) minimize $f(\hat{X}; C)$.

There are different approaches to learn $\mathcal{A}_\theta$. Our focus is unsupervised learning approaches where given a configuration $C$, *no ground-truth solution $X^*$ is used during the training*. $\theta$ can only be optimized just based on the knowledge of the cost and the constraint, or their proxies. Note that in this work, for PCO problems, the proxies are assumed to be trained based on ground-truth values of $f(X; C)$ given different $(X, C)$ pairs, which is supervised. Unsupervised learning in this work refers to the way to learn $\mathcal{A}_\theta(C)$.

**Erdős' Probabilistic Method (EPM).** The EPM has recently been brought for LCO [1]. Specifically, The EPM formulates $\mathcal{A}_\theta(C)$ as a randomized algorithm that essentially gives a probabilistic

---

[2]Normalization $(g(\cdot; C) - g_{\min})/(g_{\min}^+ - g_{\min})$ where $g_{\min}^+ = \min_{X \in \{0,1\}^n \setminus \Omega} g(X; C)$ and $g_{\min} = \min_{X \in \{0,1\}^n} g(X; C)$ always satisfies the property. $g_{\min}^+, g_{\min}$ often can be easily estimated in practice.

distribution over the solution space $\{0, 1\}^n$, which solves the optimization problem:

$$\min_\theta \quad \mathbb{E}\left[l(X; C)\right], \text{ where } l(X; C) \triangleq f(X; C) + \beta 1_{g(X;C) \geq 1}, X \sim \mathcal{A}_\theta(C) \text{ and } \beta > 0 . \quad (2)$$

Karalias & Loukas proved that with $\beta > \max_{X \in \Omega} f(X; C)$ and a small expected loss $\mathbb{E}\left[l(X, C)\right] < \beta$, sampling a sufficiently large number of $\hat{X} \sim \mathcal{A}_\theta(C)$ guarantees the existance of a feasible $\hat{X} \in \Omega$ that achieves the cost $f(\hat{X}; C) \leq \mathbb{E}\left[l(X, C)\right]$ [1]. Although this guarantee makes EPM intriguing, applying EPM in practice is non-trivial. We will explain the challenge in Sec. 3.1, which inspires our solutions and further guides the objective design for general CO and PCO problems.

## 3 The Relaxation Principle for Unsupervised LCO

In this section, we start with the practical issues of EPM. Then, we introduce our solutions by proposing a relaxation principle of the objectives, which gives performance guarantee for general practical unsupervised LCO.

### 3.1 Motivation: The Practical Issues of EPM

Applying EPM in practice has two fundamental difficulties. First, optimizing $\theta$ in Eq.(2) is generally hard as the gradient $\frac{dX}{d\theta}$ (note here each entry of $X$ is binary) may not exist so the chain rule cannot be used. We discuss the potential solutions to this problem in Sec. 3.4. Second, EPM needs to sample a large number of $X \sim \mathcal{A}_\theta(C)$ for evaluation to achieve the performance guarantee in [1]. This is not acceptable where the evaluation per sample is time-consuming and expensive.

So, in practice, Karalias & Loukas consider a deterministic method. They view $\mathcal{A}_\theta(C) \in [0, 1]^n$ as the parameters of Bernouli distributions to generate the entries of $X$ so $\mathbb{E}[X] = \mathcal{A}_\theta(C)$. First, they optimize $\min_\theta l(\mathcal{A}_\theta(C), C)$ instead of $\min_\theta \mathbb{E}[l(X, C)]$, and then, sequentially round the probability $\mathcal{A}_\theta(C)$ to discrete $X \in \{0, 1\}^n$ by comparing conditional expectations, e.g., $\mathbb{E}[l(X, C)|X_1 = 0]$ v.s. $\mathbb{E}[l(X, C)|X_1 = 1]$ to decide $X_1$. However, such conditional expectations cannot be efficiently computed unless one uses Monte-Carlo sampling or $l$ has special structures as used in the two case studies in [1], i.e., max-clique and graph-partition problems. However, what special structures are needed has not been defined, which blocks the applicability of this framework to general CO problems, especially for the PCO problems where the objectives $l$ are learned as models.

### 3.2 Our Approach: Relaxation plus Rounding, and Performance Guarantee

Our solution does not use the probabilistic modeling but directly adopts a relaxation-plus-rounding approach. We optimize a relaxation of the objective $l_r$ and obtain a soft solution $\bar{X} \in [0, 1]^n$. Then, we deterministically round the entries in $\bar{X}$ to a solution $X$ in the discrete space $\{0, 1\}^n$. Note that throughout the paper, we use $\bar{X}$ to denote a soft solution and $X$ to denote a discrete solution. The question is whether the obtained solution may still achieve the guarantee as EPM does. Our key observation is that such success essentially depends on how to relax the objective $l$.

Therefore, our first contribution beyond [1] is to propose the principle (Def. 2) to relax general costs and constraints. With this principle, the unsupervised LCO framework can deterministically yield valid and low-cost solutions (Thm. 1) as the EPM guarantees, and is applied to any objective $l$.

First, we introduce the pipeline. Consider a relaxation of a deterministic upper bound of Eq.(2):

$$\min_\theta l_r(\theta; C) \triangleq f_r(\bar{X}; C) + \beta g_r(\bar{X}; C), \text{ where } \bar{X} = \mathcal{A}_\theta(C) \in [0, 1]^n , \beta > 0. \quad (3)$$

Here $f_r(\cdot; C) : [0, 1]^n \to \mathbb{R}$ is the relaxation of $f(\cdot; C)$, which satisfies $f_r(X; C) = f(X; C)$ for $X \in \{0, 1\}^n$. The relation between the constraint $g$ and its relaxation $g_r$ is similar, i.e., $g_r(X; C) = g(X; C)$ for $X \in \{0, 1\}^n$. Here, we also use the fact that $g_r(X; C)$ provides a natural upper bound $1_{g(X;C) \geq 1} \leq g_r(X; C)$ for $X \in \{0, 1\}^n$ given the normalization of $g(X; C)$ adopted in Sec. 2.

Now, suppose the parameter $\theta$ gets optimized so that $l_r(\theta; C)$ is small. Further, we adopt the sequential rounding in Def. 1 to adjust the continuous solution $\bar{X} = \mathcal{A}_\theta(C)$ to discrete solution $X$.

**Definition 1** (Rounding). *Given a continuous vector $\bar{X} \in [0, 1]^n$ and an arbitrary order of the entries, w.o.l.g., $i = 1, 2, ..., n$, round $\bar{X}_i$ into 0 or 1 and fix all the other variables un-changed. Set $X_i = \arg\min_{j=0,1} f_r(X_1, ..., X_{i-1}, j, \bar{X}_{i+1}, ..., \bar{X}_n; C) + \beta g_r(X_1, ..., X_{i-1}, j, \bar{X}_{i+1}, ..., \bar{X}_n; C)$, replace $\bar{X}_i$ with $X_i$ and repeat the above procedure until all the variables become discrete.*

Note that our rounding procedure does not need to evaluate any conditional expectations $\mathbb{E}[l(X;C)|X_1]$ which EPM in [1] requires. Instead, we ask both relaxations $f_r$ and $g_r$ to satisfy the principle in Def. 2. With this principle, the pipeline allows achieving a valid and low-cost solution $X$, as proved in Theorem 1. We leave the proof in Appendix A.1.

**Definition 2** (The Entry-wise Concave Principle). *For any $C \in \mathcal{C}$, $h_r(\cdot; C) : [0,1]^n \to \mathbb{R}$ is entry-wise concave if for any $\gamma \in [0,1]$ and any $\bar{X}, \bar{X}' \in [0,1]^n$ that are only different in one entry,*

$$\gamma h_r(\bar{X};C) + (1-\gamma)h_r(\bar{X}';C) \le h_r(\gamma\bar{X} + (1-\gamma)\bar{X}';C).$$

Note that entry-wise concavity is much weaker than concavity. For example, the function $h_r(\bar{X}_1, \bar{X}_2) = -\text{Relu}(\bar{X}_1\bar{X}_2)$, $\bar{X}_1, \bar{X}_2 \in \mathbb{R}$ is entry-wise concave but not concave.

**Theorem 1** (Performance Guarantee). *Let $\beta > \max_{X \in \Omega} f(X;C)$ and $\min_{X \in \Omega} f(X;C) \ge 0$ in Eq.(3). Suppose the relaxed cost $f_r$ and constraint $g_r$ are entry-wise concave, and the learned parameter $\theta$ achieves $l_r(\theta;C) < \beta$. Then, rounding (Def. 1) the relaxed solution $\bar{X} = \mathcal{A}_\theta(C)$ generates a valid discrete solution $X \in \Omega$ such that $f(X;C) < l_r(\theta;C)$.*

When there are multiple constraints $g^{(j)}(X;C) < 1$ for $j = 1, 2, ...$, we may use relaxation $\beta \sum_j g_r^{(j)}(X;C)$ as the penalty term in Eq.(3), where $g_r^{(j)}$ is a relaxation of $g^{(j)}$. It can be shown that if $\sum_j g_r^{(j)}$ satisfies the condition of entry-wise concavity, the guarantee of Thm. 1 still applies.

### 3.3 The Wide Applicability of Entry-wise Concave Relaxations

We have introduced the entry-wise concave principle to relax the objective to associate our framework with performance guarantee. The question is how widely applicable this principle could be.

Actually, every function with binary inputs can be relaxed as an entry-wise affine (also called multi-linear) function with the exactly same values at the discrete inputs, as shown in Theorem 2. Note that entry-wise affinity is a special case of entry-wise concavity. In Sec. 5, we will provide the design of NN architecture (for PCO) and math derivation (for CO) that guarantee formulating an entry-wise concave function. Note that the objectives for max-clique and graph-partition problems used in [1] are essentially entry-wise affine.

**Theorem 2** (Wide Applicability). *For any binary-input function $h(\cdot) : \{0,1\}^n \to \mathbb{R}$, there exists a relaxation $h_r(\cdot) : [0,1]^n \to \mathbb{R}$ such that (a) $h_r(X) = h(X)$ for $X \in \{0,1\}^n$ and (b) $h_r$ is entry-wise affine, i.e., for any $\gamma \in [0,1]$ and any $\bar{X}, \bar{X}' \in [0,1]^n$ that are only different in one entry,*

$$\gamma h_r(\bar{X}) + (1-\gamma)h_r(\bar{X}') = h_r(\gamma\bar{X} + (1-\gamma)\bar{X}').$$

*Proof sketch.* Set $h_r(\bar{X}) = \sum_{X \in \{0,1\}^n} h(X) \prod_{j=1}^n \bar{X}_j^{X_j}(1-\bar{X}_j)^{(1-X_j)}$, which satisfies (a) and (b). Note that we suppose that $\bar{X}_j^0 = 1$ for any $\bar{X}_j \in [0,1]$. The detailed proof is in Appendix A.2. □

Although Theorem 2 shows the existence of entry-wise affine relaxations, the constructed representation in the proof depends on higher-order moments of the input entries, which make it hard to be implemented by a model, say a NN architecture. However, we claim that using entry-wise concave functions is able to implicitly generate higher-order moments via representations based on low-order moments. For example, when $n = 2$, we could use the composition of $-\text{Relu}(\cdot)$ and affine operators (only 1st-order moments) to achieve universal representation (See Prop. 1 and the proof in Appendix A.3). For general $n$, we leave as a future study.

**Proposition 1.** *For any binary-input function $h(X_1, X_2)$, there exists parameters $\{w_{ij}\}$ such that an entry-wise concave function $h_r(\bar{X}_1, \bar{X}_2) = w_{00} - \sum_{i=1}^3 \text{Relu}(w_{i1}\bar{X}_1 + w_{i2}\bar{X}_2 + w_{i0})$ satisfies $h_r(X_1, X_2) = h(X_1, X_2)$ for any $X_1, X_2 \in \{0,1\}$.*

### 3.4 Discussion: Methods to Directly Optimize the Randomized Objective in EPM Eq.(2)

The naïve way to optimize the randomized objective in Eq.(2) without worrying about the specific form of the objective $l$ is based on the policy gradient in RL via the logarithmic trick, i.e., estimating the gradient $\frac{dl}{d\theta}$ via $(f(X;C)+\beta 1_{g(X;C)\ge 1}) \log \mathbb{P}(X)$ by sampling $X \sim \mathcal{A}_\theta(C)$. However, the policy gradient suffers from notoriously large variance [39] and makes RL hard to converge. Therefore, methods such as actor critic [60] or subtracting some baselines $l(X;C) - b$ [61] have been proposed.

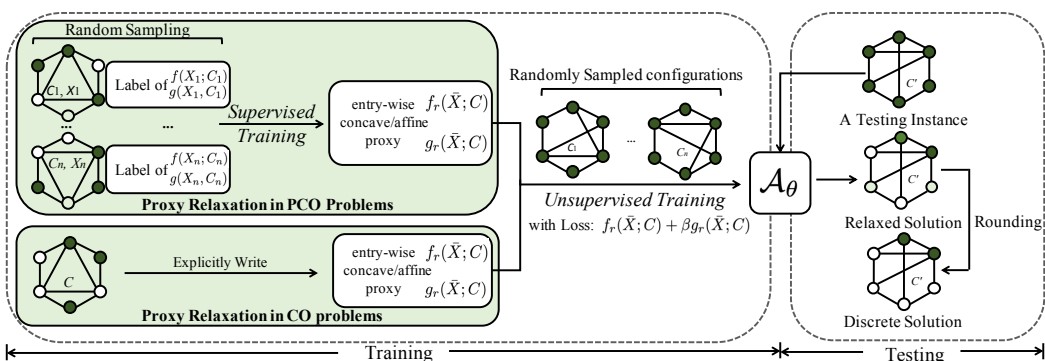

Figure 2: The pipeline of empirical evaluation in Sec. 4 and Sec. 5 on our relaxation-and-rounding principle. For a PCO task whose cost function or constraints are unknown, we first build NNs with the entry-wise concave (CON) / affine (AFF) structure to learn as their proxies ($f_r(\bar{X}; C), g_r(\bar{X}; C)$) via supervised learning. For traditional CO tasks, we follow our principle and explicitly write their entry-wise affine costs and constraints relaxation. Then, we learn the algorithm $\mathcal{A}_\theta$ to optimize the relaxed loss function $f_r(\bar{X}; C) + \beta g_r(\bar{X}; C)$ in an unsupervised manner. After training, with any unseen testing instance $C'$, we run our $\mathcal{A}_\theta$ to infer the relaxed soft solution $\bar{X}$, and then round the soft solution into discrete solution $X$ with performance guarantee.

Another way to solve Eq.(2) is based on repa-rameterization tricks to reduce the variance of gradients [62,63]. Specifically, we set the entries of output $\bar{X} = \mathcal{A}_\theta(C) \in [0, 1]^n$ as the parameters of Bernoulli distributions to generate $X$, i.e., $X_i \sim \text{Bern}(\bar{X}_i)$, for $1 \le i \le n$. To make $dX_i/d\bar{X}_i$ computable, we may use the Gumble-softmax trick [47–49]. However, this approach suffers from two issues. First, the estimation of the gradient is biased. Second, as $\mathcal{A}_\theta(C)$ is essentially a randomized algorithm,

|  | RL | Gumbel-softmax | Ours |
|---|---|---|---|
| Objective | No Limit | No Limit | Entry-wise Concave |
| Optimizer | Log Trick | Gumbel Trick | No Limit |
| Inference | Sampling | Sampling | Deter. Rounding |
| Train. Time | Slow | Fast | Fast |
| Convergence | Hard | Medium | Easy |
| Infer. Time | Slow | Slow | Fast |

Table 1: The comparison among RL (policy gradient), Gumbel-softmax methods and our principled objective relaxation. Our methods are in need of much less training time and inference time.

sampling sufficiently many $X \sim \mathcal{A}_\theta(C)$ is needed to guarantee a valid and low-cost solution. However, such evaluation is costly as discussed in Sec. 3.1. So, empirically, we can also compare $\mathcal{A}_\theta(C)$ with a threshold to determine $X$, which does not have performance guarantee. We compare different aspects of RL, Gumbel-softmax tricks and our relaxation approach in Table 1.

## 4 Applying Our Relaxation Principle to Learning for CO

First, we test our relaxation principle in a learning for CO (max clique) task, where we can explicitly write both the cost functions and the constraints into an entry-wise affine form. In this case, our framework and EPM [1] share the same pipeline, though the relaxation principle and the deterministic performance guarantee are firstly proposed in this work. The entry-wise

| Method | Twitter | RBtest |
|---|---|---|
| Badloss+R | 0.768 ± 0.203 (0.17s/g) | 0.702 ± 0.102 (0.33s/g) |
| EPM [1] | 0.924 ± 0.133 (0.17s/g) | 0.788 ± 0.065 (0.23s/g) |
| AFF (ours) | 0.926 ± 0.113 (0.17s/g) | 0.787 ± 0.065 (0.33s/g) |

Table 2: Approximation Rate in the max clique. 's/g' denotes the average time cost per graph.

affine objective relaxation of the max clique is $-(\beta + 1) \sum_{(i,j) \in E} \bar{X}_i \bar{X}_j + \frac{\beta}{2} \sum_{i \neq j} \bar{X}_i \bar{X}_j$. Here, we use a real-world dataset (Twitter [64]) and a synthetic dataset (RBtest [1]), and show the experiment results in Table. 2. We follow the settings in [1] and use a 6:2:2 dataset split for training, validating and testing, each test instance runs within 8 seeds. Our entry-wise affine pipeline achieves almost the same performance as EPM. To show the importance of the relaxation principle, we also propose 'Badloss+R'. This baseline imposes trigonometric functions to the original entry-wise affine functions $f'_r(\bar{X}; C) = f_r(\sin(9\pi\bar{X}/2); C), g'_r(\bar{X}; C) = g_r(\sin(9\pi\bar{X}/2); C)$, where $\sin(\cdot)$ operates on each entry of the input vector. Note that the relaxed functions also match the original objectives at discrete points, i.e., $f'_r(X; C) = f_r(X; C) = f(X; C)$ when $X \in \{0, 1\}$, while with different relaxations. The poor performance of 'Badloss+R' reveals the importance of our principle for relaxation.

# 5 Applying Our Relaxation Principle to Learning for PCO

In this section, we apply our relaxation principle to three PCO applications: (I) feature-based edge covering & node matching, (II) resource allocation in circuit design, and (III) imprecise functional unit assignment in approximate computing. All the applications have graph-based configurations $C$. So later, we first introduce how to use graph neural networks (GNNs) to build proxies that satisfy our relaxation principle. Such GNN-based proxies will be used as the cost function relaxation $f_r$ in all the applications. Our principle can also guide the relaxation of explicit CO objectives. The constraints in applications (I)(III) are explicit and their relaxation can be written into the entry-wise affine form. The constraint in (II) needs another GNN-based entry-wise concave proxy to learn.

## 5.1 GNN-based Entry-wise Concave Proxies

We consider the data configuration $C$ as an attributed graph $(V, E, Z)$ where $V$ is the node set, $E \subseteq V \times V$ is the edge set and $Z$ is the node attributes. We associate each node with a binary variable and group them together $X :\in \{0,1\}^{|V|}$. where for each $v \in V$, $X_v = 1$ indicates the choice of the node $v$. Note that our approach can be similarly applied to edge-level variables (see Appendix C.3), which is used in application (I). Let $\bar{X}$ still denote the relaxation of $X$.

To learn a discrete function $h : \{0,1\}^{|V|} \times C \to \mathbb{R}$, we adopt a GNN as the relaxed proxy of $h$. We first define a latent graph representation in $\mathbb{R}^F$ whose entries are all entry-wise affine mappings of $X$.

$$\textbf{Latent representation:} \qquad \phi(\bar{X}; C) = W + \sum_{v \in V} U_v \bar{X}_1 + \sum_{v,u \in V, (v,u) \in E} Q_{v,u} \bar{X}_v \bar{X}_u \qquad (4)$$

where $W$ is the graph representation, $U_v$'s are node representations and $Q_{v,u}$ are edge representations. These representations do not contain $X$ and are given by GNN encoding $C$. Here, we consider at most 2nd-order moments based on adjacent nodes as they can be easily implemented via current GNN platforms [65, 66]. Then, we use $\phi$ to generate entry-wise affine & concave proxies as follows.

$$\textbf{Entry-wise Affine Proxy (AFF):} \qquad h_r^a(\bar{X}; C) = \langle w^a, \phi(\bar{X}; C) \rangle. \qquad (5)$$

$$\textbf{Entry-wise Concave Proxy (CON):} \qquad h_r^c(\bar{X}; C) = \langle w^c, -\text{ReLU}(\phi(\bar{X}; C)) \rangle + b. \qquad (6)$$

where $w^a, w^c \in \mathbb{R}^F, b \in \mathbb{R}$ are learnt parameters and $w^c \geq 0$ guarantees entry-wise concavity. Other ways to implement GNN-based entry-wise concave proxies are also introduced in Appendix C.1.

## 5.2 The Setting up of the Experiments

**Training & Evaluation Pipeline.** In all the applications, we adopt the following training & evaluation pipeline. First, we have a set of observed configurations $\mathcal{D}_1 \subset \mathcal{C}$. Each $C \in \mathcal{D}_1$ is paired with one $X \in \{0,1\}^n$. We use the costs $f(X, C)$ (and constraints $g(X, C)$) to train the relaxed proxies $f_r(X, C)$ (and $g_r(X, C)$, if cannot be derived explicitly), where the relaxed proxies follow either Eq.(5) (named AFF) or Eq.(6) (named CON). Then, we parameterize the LCO algorithm $\mathcal{A}_\theta(C) \in$

| Baseline | $f_r, g_r$ | $\mathcal{A}_\theta$ | Inference |
|---|---|---|---|
| Naïve + R | no limit | no limit | rounding |
| RL | no limit | RL | sampling |
| GS-Tr+S | no limit | GS | sampling |
| GS-Tr+R | no limit | GS | rounding |

Table 3: The baselines in the paper.

$[0, 1]^n$ via another GNN. Based on the learned (or derived) $f_r$ and $g_r$, we optimize $\theta$ by minimizing $\sum_{C \in \mathcal{D}_1} l_r(\theta; C)$, where $l_r$ is defined according to Eq.(3). We will split $\mathcal{D}_1$ into a training set and a validation set for hyperparameter-tuning of both proxies and $\mathcal{A}_\theta$. We have another set of configurations $\mathcal{D}_2 \subset \mathcal{C}$ used for testing. For each $C \in \mathcal{D}_2$, we use the relaxation $\bar{X} = \mathcal{A}_\theta(C)$ plus our rounding to evaluate the learned algorithm $\mathcal{A}_\theta(\cdot)$. We follow [1] and do not consider fine-tuning $\mathcal{A}_\theta$ over the testing dataset $\mathcal{D}_2$ to match the potential requirement of the fast inference.

**Baselines.** We consider 4 common baselines that is made up of different learnable relaxed proxies $f_r, g_r$, algorithms $\mathcal{A}_\theta$ and inference approaches as shown in Table 3. For the proxies $f_r, g_r$ for baselines, we apply GNNs without the entry-wise concave constraint and use $X$ as one node attribute while keeping all other hyper-parameters exactly the same as CON other than the way to deal with the discrete variables to make fair comparison (See details in Appendix. C); For the algorithm $\mathcal{A}_\theta$, we provide the Gumbel-softmax trick based methods (GS-Tr) [48, 49], the actor-critic-based RL method [60] (RL) and the naïve relaxation method (Naïve); For the inference approaches, we consider Monte Carlo sampling (S) and our proposed rounding (R) procedure. Although the baselines adopt proxies that are different from ours, we guarantee that their proxies approximate the ground-truth

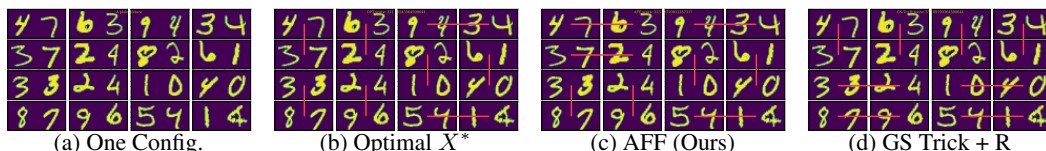

| (a) One Config. | (b) Optimal $X^*$ | (c) AFF (Ours) | (d) GS Trick + R |

Figure 3: The visualization for node matching in Application I. Our method avoids large multiplications $87 * 96$ and $94 * 82$ where GS-Trick cannot, and generate a solution different but close to OPT.

$f, g$ over the validation dataset at least no worse than ours. In application II, we also consider two non-learnable algorithms to optimize the proxies without relaxation constraints, simulated annealing (SA) [35] and genetic algorithms (GA) [67, 68]. In application III, we put all of the required AxC units either close to the input (C-In) or close to the output (C-Out) of the approximating computing circuit as additional baselines. More details of the experiments setups and hyperparameter tuning can be found in Appendix C. We also obtain the optimal solutions (OPT) for applications I and III via brute-force search for comparison.

### 5.3 Application I: Feature-based Edge Covering & Node Matching in Graphs

This application is inspired by [69]. Here, each configuration $C$ is a $4 \times 4$ grid graph whose node attributes are two-digit images generated by random combinations of the pictures in MNIST [70]. We associated each edge with variables $X \in \{0, 1\}^{|E|}$. The objective is the sum of edge weights $f(X; C) = \sum_{e \in E} w_e X_e$ where $w_e$ is unknown in prior and needed to be learned. The ground truth of $w_e$ is a multiplication of the numbers indicated by the images on the two adjacent nodes. We adopt ResNet-50 [71] (to refine node features) plus GraphSAGE [72] to encode $C$. We consider using both Eq.(5) and Eq.(6) to formulate the relaxed cost $f_r(\bar{X}; C)$. Training and validating $f_r$ are based on 100k randomly sampled $C$ paired with randomly sampled $X$. Note that 100k is much smaller than the entire space $\{0, 1\}^{|E|} \times \mathcal{C}$ is of size $2^{24} \times 100^{16}$.

Next, as the constraint here is explicit, we can derive the relaxation of the constraints for this application. First, the constraint relaxation of the edge covering problem can be written as

$$\textbf{Edge Covering Constraint:} \quad g_r(\bar{X}; C) = \sum_{v \in V} \prod_{e:v \in e} (1 - \bar{X}_e). \tag{7}$$

Each production term in Eq.(7) indicates that for each node, at least one edge is selected. We can easily justify that $g_r$ is entry-wise affine and $\Omega = \{X \in \{0, 1\}^{|E|} : g_r(X; C) < 1\}$ exactly gives the feasible solutions to the edge covering problem.

Similarly, we can derive the constraint for node matching by adding a further term to Eq.(7).

$$\textbf{Node Matching Constraint:} \; g_r(\bar{X}; C) = \sum_{v \in V} [\prod_{e:v \in e} (1 - \bar{X}_e) + \prod_{e_1, e_2:v \in e_1, e_2, e_1 \neq e_2} \bar{X}_{e_1} \bar{X}_{e_2}]. \tag{8}$$

Here, the second term indicates that no two edges adjacent to the same node can be selected. This is a case with two constraints while we combine them together. We can easily justify that $g_r$ is entry-wise affine and $\Omega = \{X \in \{0, 1\}^{|E|} : g_r(X; C) < 1\}$ groups exactly the feasible solutions to the node matching problem.

Note that our above derivation also generalizes the node-selection framework in [1] to edge selection. With the learned $f_r$ and the derived $g_r$, we further train and validate $\mathcal{A}_\theta$ over the 100k sampled $(X, C)$'s and test on another 500 randomly sampled $C$'s.

**Evaluation.** Table 4 shows the evaluation results. In the GS-Tr+S method, the number of sampling is set to 120 (about 2.5 times the inference time of our deterministic rounding). Note that for node matching, GS-Tr+S could hardly sample a feasible node matching solution within 120 samples. The experiment results show that our principled proxy relaxation exceeds the other baselines on both tasks. Also, we observe that AFF outperforms CON, which results from the fact that $f(X; C)$ in these two problems are naturally in entry-wise affine forms

| Method | Edge covering | Node matching |
|---|---|---|
| Naive+R | 68.52 | 429.12 |
| RL | 51.29 | 426.97 |
| GS-Tr+S | 63.36 | - |
| GS-Tr+R | 46.91 | 429.39 |
| CON(ours) | 49.59 | 422.47 |
| AFF(ours) | **44.55** | **418.96** |
| OPT(gt) | 42.69 | 416.01 |

Table 4: Performance on application I (graph optimization).

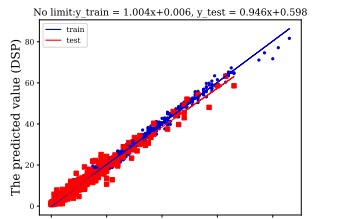 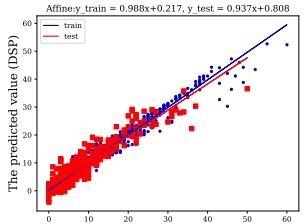 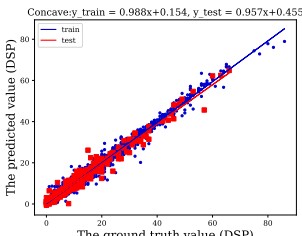

Figure 4: Comparing different proxies for learning DSP usage. Left, no constraint; Middle, entry-wise affine constraint (Eq. (5)); Right, entry-wise concave constraint (Eq.(6))

with low-order (1st-order) moments. One instance of node matching randomly selected from the test set is shown in Fig. 3. More visualization results can be found in Fig. 6 in the appendix.

## 5.4 Application II: Resource Allocation in Circuit Design

Resource allocation in field-programmable gate array (FPGA) design is a fundamental problem which can lead to largely varied circuit quality after synthesis, such as area, timing, and latency. In this application, we follow the problem formulation in [24, 30], where the circuit is represented as a data flow graph (DFG), and each node represents an arithmetic operation such as multiplication or addition. The goal is to find a resource allocation for each node to be either digital signal processor (DSP) or look-up table (LUT), such that the final circuit area (i.e., actual DSP and LUT usage) after synthesis is minimized. Notably, different allocation solutions result in greatly varied DSP/LUT usage due to complicated synthesis process, which cannot be simply summed up over each node. To obtain precise DSP/LUT usage, one must run high-level synthesis (HLS) [73] and place-and-route [74] tools, which can take up to hours [24, 30].

In this application, each configuration $C$ is a DFG with $> 100$ nodes, where each node is allocated to either DSP or LUT. Node attributes include operation type (i.e., multiplication or addition) and data bitwidth. More details about the dataset can be found in Appendix C.5. Let $X \in \{0, 1\}^{|V|}$ denote the mapping to DSP or LUT. Let $f_r$ and $g_r$ denote the proxies of actual LUT and actual DSP usage, respectively. Note that given different constraints on the DSP usage, we will normalize $g_r$ as introduced in Sec. 2. We train and validate $f_r, g_r, \mathcal{A}_\theta$ on $8,000$ instances that consist of $40$ DFGs ($C$), each DFG with $200$ different mappings ($X$), and test $\mathcal{A}_\theta$ over $20$ DFGs. Note that the actual LUT and DSP usages of each training instance has been collected by running HLS in prior. We also run HLS to evaluate the actual LUT and DSP usages for the testing cases given the learned mappings.

**Evaluation.** We rank each method's best actual LUT usage under the constraint of different percentages ($40\%$ - $70\%$) of the maximum DSP usage in each testing instance, then calculate the averaged ranks. Fig. 5 shows the results. Our entry-wise concave proxy achieves the best performance. GS-Tr+R is slightly better than RL, and both of them exceed SA and GA. We do not include our entry-wise affine proxy in the ranking list, because the affine proxy could be much less accurate than the proxy without constraints and the entry-wise concave proxy. The comparison between these proxies on learning DSP usage (& LUT usage) is shown in Fig. 4 (& Fig. 9 in the appendix, respectively). The gap between different proxies indicates the FPGA circuit contains high-order moments of the input optimization variables and 2-order entry-wise affine proxy cannot model well. We do not include the result of GS-Tr+S and Naive+R, because these methods perform poor and could hardly generate feasible solutions given a constraint of DSP usage. We leave their results in Table. 7 in the appendix. Moreover, we compare the training time between different methods. To be fair, all methods run on the same server with a Quadro RTX 6000 GPU. The RL based optimizer takes 22 GB GRAM, while other optimizers only take 7 GB on average. Fig. 10 in the appendix further demonstrates that our methods and GS-T methods require much less training time than RL.

Also, to give a fair comparison between learning-based approaches and traditional approaches, we implement GA with parallel (on GPU) cost-value inference for all the populations in each generation. We set the population size as $256$, which is the same as the batch size that we used to train/infer $\mathcal{A}_\theta$. The performance of GA in Fig. 5 is obtained under the condition that the inference time of the implemented parallel GA is about the same as that of our CON method. Fig. 8 in the appendix provides more detailed comparison on the performance and the inference time between GA with different numbers of generations and our CON method.

| DSP usage | 40% | 45% | 50% | 55% | 60% | 65% | 70% | rank-avg |
|---|---|---|---|---|---|---|---|---|
| SA | 3.41 | 3.08 | 3.50 | 3.33 | 3.66 | 4.16 | 4.08 | 3.60 |
| GA | 2.75 | 3.41 | 2.83 | 2.91 | 3.00 | 3.00 | 2.75 | 2.95 |
| RL | 3.33 | 3.58 | 3.83 | 3.25 | 2.91 | 2.83 | **2.41** | 3.16 |
| GS-Tr+R | 3.58 | 2.91 | 2.58 | 3.16 | **2.33** | **2.58** | 3.00 | 2.87 |
| CON | **1.83** | **2.00** | **2.25** | **2.25** | 3.08 | 2.41 | 2.66 | **2.35** |

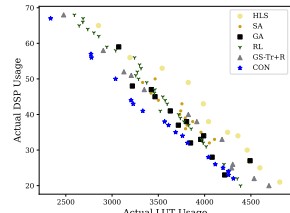

Figure 5: The left table shows averaged ranks of the LUT usage given by different methods with the constraint of different percentage of DSP usage in Application II (resource allocation). The right figure shows the DSP-LUT usage amount relationship on one test configuration. The HLS baseline denotes the optimal synthesis results among 200 random mappings.

| Threshold $\theta$ | C-In | C-Out | Naïve | RL | GS-Tr+S | GS-Tr+R | CON | AFF | OPT |
|---|---|---|---|---|---|---|---|---|---|
| 3 AxC units | 12.42 | 12.44 | 3.62 | 7.68 | 4.87 | 3.24 | 3.18 | **3.10** | 2.77 |
| 5 AxC units | 14.68 | 14.65 | 6.20 | 10.15 | 8.03 | 5.86 | **5.13** | 5.38 | 4.74 |
| 8 AxC units | 17.07 | 17.04 | 11.12 | 12.83 | 12.65 | 10.62 | 10.17 | **10.04** | 8.56 |

Table 5: Relative errors of different methods with the AxC unit constraint as 3,5,8 in Application III.

### 5.5 Application III: Imprecise Functional Unit Assignment in Approximate Computing

One fundamental problem in approximate computing (AxC) is to assign imprecise functional units (a.k.a., AxC units) to execute operations such as multiplication or addition [42–46], aiming to significantly reduce circuit energy with tolerable error. We follow the problem formulation in [45], where given a computation graph, each node represents either multiplication or addition. The incoming edges of a node represent its two operands. The goal is to assign AxC units to a certain number of nodes while minimizing the expected relative error of the output of the computation graph.

In this application, each configuration $C$ is a computation graph with 15 nodes (either multiplication or addition) that maps a vector in $\mathbb{R}^{16}$ to $\mathbb{R}$. A fixed number $\theta$ of nodes are assigned to AxC units with produce 10% relative error. Let $X \in \{0, 1\}^{|V|}$ denote whether a node is assigned to an AxC unit or not; the proxy of the objective $f_r$ is the expected relative error at the output. We use 100k $(X, C)$ as the training dataset and the entire solution space is $2^{15} \times 2^{15}$. For each $(X, C)$, the ground-truth, i.e., expected relative error, is computed by averaging 1k inputs sampled uniformly at random from $[0, 10]^{16}$. The constraint $g_r$ is $\sum_{v \in V} X_v \geq \theta$ with normalization, where $\theta \in \{3, 5, 8\}$. We test the learned $\mathcal{A}_\theta$ on 500 unseen configurations.

**Evaluation.** Table. 5 shows the averaged relative errors of the assignments by different methods. The problem is far from trivial. Intuitively, assigning AxC units closed to the output, we may expect small error. However, C-Out performs bad. Our proxies AFF and CON obtain comparable best results. The MAE loss values of the two proxies are also similar, as shown in Table 9 in the appendix. The reason is that the circuit is made up of 4 layers in total which leads to at most 4-order moments in the objective function, which is in a medium-level complexity. Training time is also studied for this application, resulting in the same conclusion as application II (See Table 8 in the appendix).

## 6 Conclusion

This work introduces an unsupervised end-to-end framework to resolve LCO problems based on the relaxation-plus-rounding technique. With our entry-wise concave architecture, our framework guarantees that a low objective value could lead to qualified discrete solutions. Our framework is particularly good at solving PCO problems where the objectives need to be modeled and learned. Real-world applications demonstrate the superiority of our method over RL and gradient-relaxation approaches in both optimization performance and training efficiency. In the future, we aim to further broaden our framework to the applications where the optimization variables allow using more complex embeddings than binary embeddings.

## 7 Acknowledgement

We greatly thank all the reviewers for their valuable feedback and the insightful suggestions. H. Wang and P. Li are partially supported by 2021 JPMorgan Faculty Award and the NSF award OAC-2117997.

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
