# OpenReview forum: "Unsupervised Learning for Combinatorial Optimization with Principled Objective Relaxation"
_NeurIPS.cc/2022/Conference — NeurIPS 2022 Accept_

### Official Review · Reviewer_kjHC · 2022-07-08

**Rating:** 4
**Confidence:** 4
**Soundness:** 1 poor
**Presentation:** 3 good
**Contribution:** 3 good

**Summary:**

This paper extends the relax-optimize-discretize pipeline in Erdos Goes Neural. The authors believe that the sequential decoding scheme in EGN suffers from two issues, and resolve these issues by sequentially rounding over a so-called entry-wise concave principle. The proposed scheme is claimed to require fewer samples and thus faster than EGN, while still preserving the theoretical performance guarantee in EGN. The authors mainly investigate the application of the proposed approach in proxy combinatorial optimization (PCO), where the objective function is unknown or too expansive to know.

**Questions:**

1. Is there any reason why pure CO problems are not evaluated?
2. Are the proxy-GNNs learned by supervised learning?
3. In application II, what is the proxy model for RL/SA/GA? Do they use the same entry-wise concave GNN? It seems that we can build a conventional GNN for them and I am curious whether a conventional GNN will be more accurate?

**Limitations:**

The authors write in the checklist that the limitations are addressed in Section 5, but it is missing.

**Strengths And Weaknesses:**

Here is my overall judgment of this paper: the ideas/insights/experiments are novel and interesting. However, this paper reads somewhat confusing, especially since the experiments seem not to match the other parts of this paper.

**Strengths**
1. (Unsupervised) learning for combinatorial optimization is an important topic worth studying. The considered PCO problem is also practical and important.
1. I enjoy the narrative of theoretical analysis and insights about learning CO.


**Weakness**
1. Experiment for learning on pure (non-proxy) CO problems is missing, and the important baseline Erdos Goes Neural is missing. In the experiment, the authors study 3 PCO problems, which I agree to be very important and novel. However, in Section 1&2&3, this paper reads to be an improved version of Erdos Goes Neural. Since Erdos Goes Neural is focused on pure CO problems, there is no reason not to evaluate such problems and compare to EGN to support the theoretical results. Besides, the authors also discussed on how to tackle the pure CO problem in detail in Section 3. It seems confused for proposing a new method for pure CO but evaluating it.
2. This paper is titled "unsupervised" learning. However, it seems that the evaluated PCO problems are not "unsupervised", but "supervised" because the proxy should be learned from labels. For example, in application II, the authors write "the actual LUT and DSP usages of each training instance has been collected by running HLS in prior".
3. Adding a flow chart or an algorithm block for the proposed framework will make this paper clearer and easier to reimplement.
4. The implementation details on the application of PCO problems seem not clear. For example: In application II, what is the proxy model for RL/SA/GA? Do they use the same entry-wise concave GNN? It seems that we can build a conventional GNN for them and I am curious whether a conventional GNN will be more accurate?


**Other Remarks**
1. L19: Most CO problems are NP-complete -> Most CO problems are NP-complete or NP-hard.
2. Is Erdos Goes Neural missing in the discussion in Table 1? How is EGN considering these aspects?
3. I feel confused about L188: "Optimizing $\theta$ in Eq.(2) is generally hard as the gradient $dX/d\theta$ does not generally exist so the chain rule cannot be used." However, $X$ should be continuous in EGN, therefore the gradient should always exist.
4. In L128: "The two special cases in [1] on the max-clique and graph-partition problems seem to have special structures." Do you have any support for this claim?

---

> ### Author Response · Authors · 2022-08-02
> **Response to reviewer kjHC (1/2)**
>
> Thank you so much for reviewing our paper, and providing very detailed comments and actionable suggestions. We feel sorry for the caused confusions. Next, we try our best to resolve other confusions that the reviewer raised.
>
> > Experiment for learning on pure (non-proxy) CO problems is missing, and the important baseline Erdos Goes Neural [1] is missing.
>
> The reason we did not compare against the most relevant work [1] based on the benchmarks was explained in the above unified response. We add some comparison on pure (non-proxy-based) CO problems, edge covering and node matching, as shown in the unified response. For our experiments in Sec. 4, as these experiments are all proxy-based CO, [1] cannot be applied.
>
> *However, we respectfully disagree with the reviewer’s statement that the experiments do not match the other parts of this paper.* We argue that our theory in Sec. 3 is applied to general CO problems including both pure CO and proxy-based CO problems. Moreover, we see the exclusive significance of this work in guiding the design of the proxy for PCO problems. Therefore, we emphasize the relation to PCO problems throughout the paper and further do experiments on PCO problems in Sec. 4.
>
> > The evaluated PCO problems are not "unsupervised", but "supervised"
>
> We are sorry about the confusion. We answer this question in the above unified response. Specifically, we say “unsupervised learning“ to refer to the procedure of learning for CO, which follows the common terminology of learning for CO. The way to learn proxy models is indeed based on supervised learning. We will make this point very clear in the final version.
>
>
> > In application II, what is the proxy model for RL/SA/GA?
>
> The reviewer’s insight behind this question is correct. For the RL/SA/GA methods,  there are no reasons to use entry-wise concave GNNs because these methods suppose to work on discrete assignments and will not touch the relaxed regime of the functions. Therefore, we adopt the GNN structure ‘with no limit’ as the proxy (i.e., the conventional GNN structure) for these baselines. We explain such details in Section 4.2 Baseline and Appendix C.
>
> > Adding a flow chart or an algorithm block for the proposed framework
>
> This is a very actionable suggestion and we greatly appreciate it. In the final version, we will have one extra page and will add a flow chart to better illustrate our framework.
>
> Our framework is as follows: Given a PCO problem, when the cost or the constraint does not have a close form, we first train CON/AFF via supervised learning to learn a cost proxy model or a constraint proxy model. Then, we do unsupervised learning to learn an NN-parameterized algorithm $A_{\theta}$ that optimizes the proxy objective. When testing, we use $A_{\theta}$ to generate a soft solution and then round the solution into a discrete solution.
>
> Given a (non-proxy-based) CO problem, as the cost and the constraint are known, we do not need CON/AFF to learn a cost proxy model or a constraint proxy model. Instead, we derive the continuous relaxations of the cost and the constraint so that the obtained relaxations are entry-wise concave. The remaining parts to learn and to test  $A_{\theta}$ are the same as PCO problems.

---

> > ### Author Response · Authors · 2022-08-02
> > **Response to reviewer kjHC (2/2)**
> >
> > > Other remarks: “ X in EGN should be continuous, therefore the gradient dX/d\theta should always exist; The two special cases in EGN have special structures; Erdos Goes Neural [1] is missing in the discussion in Table 1.”
> >
> > We find these remarks are related. So, we respond to them together.
> >
> > The relation between EGN and ours has been explained in the above unified response. A check on the unified response on this relation is helpful to understand our following response.
> >
> > We first would like to clarify our notation used in the paper. Throughout the paper, we always use $X$ without bar to denote discrete variables and $\bar{X}$ with bar to denote continuous variables. So, our statement “$dX/d\theta$ is not computable” is correct. This is an argument for the case that tries to apply the theory of EGN to general CO problems, which always encounters such non-differentiable difficulty.  For the specific cases studied in EGN, EGN adopts “$d\bar{X}/d\theta$” (not $dX/d\theta$), which essentially follows a relaxation as this paper suggests. However, EGN does not derive the relaxation principle as ours, although the used relaxation happens to satisfy our principle.
> >
> > The above description implies a gap between EGN’s theory for general CO problems and its implementation for the two cases studied in [1]. In the EGN paper, the theory is only in the probabilistic sense and works for just discrete objectives (see Eq.(3) and Thm 1 in [1]).  However, the implementation in EGN for the max-clique and graph-partition problems does not strictly follow the probabilistic model but relaxes discrete variables $X$ into their continuous counterparts $\bar{X}$. Such replacement cannot achieve the performance guarantee claimed by EGN for general CO problems, because the expectation in general does not always equal to the relaxation.
> >
> > Because of the above gap of EGN for general CO problems, we think it is tricky to well position EGN in Table 1. If we strictly follow the probabilistic description used in EGN, EGN does not specify a solution to the non-differentiable issue $dX/d\theta$ and suffers from the sampling-based costly evaluation. If we follow its case study by replacing discrete variables $X$ with their continuous counterparts $\bar{X}$, EGN does not give the rule to guarantee such replacement to have the performance guarantee for general CO problems. The derivation in the EGN appendix is specialized to the max-clique and graph-partition problems to support these two special cases. Our paper instead gives the principle for general CO problems. Even for proxy-based CO problems where the objectives are unknown, our method can be applied.
> >
> > [1]Karalias et al. Erdos goes neural: an unsupervised learning framework for combinatorial optimization on graphs. NeurIPS 2020

---

> > > ### Comment · Reviewer_kjHC · 2022-08-06
> > > **Thanks for the feedback**
> > >
> > > Thank the authors for the detailed feedback. After reading the response, I still feel confused about the following points:
> > > * Since your method degenerates to EGN when solving pure CO problems, does it mean that EGN also has the concave property as your method? If not, why do EGN and the proposed method share the same pipeline?
> > > * It seems that EGN will be very similar to the proposed method if EGN is also included in Table 1.
> > > * Other names for proxy-CO in the research community are "predict-and-optimize", "decision-focused learning", and "learning to differentiate (of non-differentiable solvers)". If proxy-CO problems are really the focus of this paper, many important baselines are missing in experiments. Just to name a few:
> > >   * Decision-Focused Learning without Decision-Making: Learning Locally Optimized Decision Losses
> > >   * Contrastive Losses and Solution Caching for Predict-and-Optimize
> > >   * Decision-Focused Learning: Through The Lens of Learning to Rank
> > >   * Differentiation of Blackbox Combinatorial Solvers
> > >   * Learning with Differentiable Perturbed Optimizers
> > >   * CombOptNet: Fit the Right NP-Hard Problem by Learning Integer Programming Constraints
> > >
> > > I truly appreciate the efforts of the authors, and I look forward to your further feedback.

---

> > > > ### Author Response · Authors · 2022-08-08
> > > > **Further response to reviewer kjHC (1/2)**
> > > >
> > > > We thank Reviewer KjHC for reading our response carefully and raising further questions. Our response to these questions are as follows.
> > > >
> > > > > “Since your method degenerates to EGN when solving pure CO problems, does it mean that EGN also has the concave property as your method? If not, why do EGN and the proposed method share the same pipeline?”
> > > >
> > > > It is incorrect to say that our method degenerates to EGN when solving pure CO problems. It is correct to say that our method degenerates to EGN **given a relaxation objective of pure CO problems that satisfy our proposed principle**. The key point here is that the paper of EGN has provided no statements on the principle of how to relax the objective. Our paper is the first work to propose such a principle.
> > > >
> > > > Note that this principle is super important, because an arbitrary way to relax the objective will lead to very bad performance. As shown in our pure CO experiments (in the unified response), the baseline **Badloss** in the experiments shows that how bad the performance will be based on an objective relaxation that violates our principle (This bad relaxed objective keeps the same as the original objective at discrete points while the relaxed regime does not follow our entry-wise concave principle). A relaxation that satisfies our proposed principle is the key to have a performance guarantee, while the paper of EGN does not provide such a performance guarantee.
> > > >
> > > > > “It seems that EGN will be very similar to the proposed method if EGN is also included in Table 1.”
> > > >
> > > > Because of the above statement, this statement is also incorrect. The reviewer seems to assume that EGN also adopts relaxation-plus-rounding by saying that EGN is very similar to our method when included in Table 1. If so, we should say EGN has no performance guarantee when being put in Table 1. The performance guarantee given in the paper of EGN is for probabilistic formulation but not for relaxation-plus-rounding approaches. The key to achieve the performance guarantee for relaxation-plus-rounding approaches is our proposed principle.

---

> > > > > ### Author Response · Authors · 2022-08-08
> > > > > **Further response to reviewer kjHC (2/2)**
> > > > >
> > > > > > If proxy-CO is the focus of this paper, there are many important baselines being missing in experiments.
> > > > >
> > > > > Our main focus is to propose a theoretically-sound principle for relaxation-plus-rounding approaches for learning for general CO problems (including both non-proxy-CO and proxy-CO). Our experiments focus on those most significant applications of our principle, i.e., proxy-CO. So, we kindly remind the reviewer not to overlook the main contribution/focus of this work.
> > > > >
> > > > > Even just for proxy-CO, we have carefully checked the papers the reviewer referred to. To the best of our understanding, none of them are valid baselines of this work. We detail the reasons as follows.
> > > > >
> > > > > $\bullet$ First, the references [1,2,3,4,5] mentioned by the reviewer are solving a problem entirely different from our work.
> > > > >
> > > > > The references [1,2,3] mostly follow a concept named ‘decision-focused learning’ [6]. In both decision-focused learning and black-box optimization [4,5], the research goal is **to learn a model that maps data features to the parameters of a given CO problem**. There will be no learning-based solvers for CO problems generated by these works [1,2,3,4,5,6]. Their CO problems are generally solved by canonical solvers such as Gurobi. However, the research goal of our work is **to learn solvers to solve CO problems**. These are two fundamentally different problems.
> > > > >
> > > > > Because of the above difference in research problems, the experimental settings are entirely different. We agree that the references [1,2,3,4,5,6] and our proxy-CO experiments are related in the sense that how data features determine the CO objective is unknown. However, the key difference is that  [1,2,3,4,6] assume that there is a known explicit CO objective, although the parameters of the CO problem are unknown, such as knowing that the objective is to achieve the minimal sum of edge weights with node matching as the constraint, while not knowing that how the edge weights depend on the end-node features. [5] does not assume such an explicit CO, but [5] relies on a collection of optimal solutions to such a CO problem with different configurations for training, which is typically inaccessible unless the CO problem is explicit. Our proxy-CO experiments assume that we neither know the explicit CO problem (in Application I, our model does not even know that the cost is the sum of edge weights though the node matching constraint can be derived by hand) nor its optimal solution for any configuration. Such a difference is very obvious for our other two applications on circuit design, as there are neither explicit CO formulations nor any accessible optimal solutions, which means that the methods in [1,2,3,4,5,6] cannot be applied at all.
> > > > >
> > > > > Mathematically, using our notations, we are to learn a CO solver $\mathcal{A}_{\theta}(C)$ given a configuration $C$ with data features. We assume that we know nothing about the objective $f(X;C)$ except some sampled values $\{f(X_i;C_i)\}$ for $i=1,2,...$. However, the works [1,2,3,4,6] assume there is an explicit CO objective $f(\cdot ;Q)$ and the work [5] is to learn an integer programming also denoted as $f(\cdot ;Q)$ w.o.l.g..  Their goals are all to learn from the mapping from original data $C$ to $Q$. Once $Q$ is learned, the problem $f(\cdot ;Q)$ can be optimized by non-learning-based methods. The works [1,2,3,6] further assume some ground-truth labels are available to learn the mapping from $C$ to $Q$, the work [4] assumes a solver is avaiable to the explicit CO objective $f(\cdot ;Q)$, while the work [5] assumes a collection of \emph{optimal solutions} of $f( \cdot ;C)$ for different $C$’s. Our proxy-CO setting does not have any access to such information.
> > > > >
> > > > > $\bullet$ Second, the reference [7] mentioned by the reviewer is actually our baseline line Gs-Tr. The random perturbation in [7] when running on binary embeddings (0 or 1) reduces to the Gumbel-softmax method we compared with, though the goal of reference [7] is to generalize Gumbel-max method to more complicated combinatorial objectives.
> > > > >
> > > > > [1]Shah et al. Decision-Focused Learning without Decision-Making: Learning Locally Optimized Decision Losses. arXiv:2203.16067
> > > > >
> > > > > [2]Mulamba et al. Contrastive Losses and Solution Caching for Predict-and-Optimize. IJCAI 2021: 2833-2840
> > > > >
> > > > > [3]Mandi et al. Decision-Focused Learning: Through The Lens of Learning to Rank. ICML 2022: 14935-14947
> > > > >
> > > > > [4]Pogancic et al. Differentiation of Blackbox Combinatorial Solvers. ICLR 2020
> > > > >
> > > > > [5]Paulus et al. CombOptNet: Fit the Right NP-Hard Problem by Learning Integer Programming Constraints. ICML 2021: 8443-8453
> > > > >
> > > > > [6]Wilder et al. Melding the Data-Decisions Pipeline: Decision-Focused Learning for Combinatorial Optimization. AAAI 2019: 1658-1665
> > > > >
> > > > > [7]Berthet et al. Learning with Differentiable Perturbed Optimizers. Neurips 2020

---

> > > > > > ### Comment · Reviewer_kjHC · 2022-08-08
> > > > > > **Thanks for your feedback**
> > > > > >
> > > > > > Thanks for the feedback, while I think there might be certain room for argument about whether my understandings of this paper are absolutely "incorrect", e.g. a probably stronger baseline for proxy-CO might be making SA/GA (or other solvers) differentiable by the black-box trick (Pogancic et al. ICLR 2020), and then training the proxy model end-to-end.
> > > > > >
> > > > > > I will retain my original score at this time and I will be happy to discuss it with other reviewers in the coming week.

---

> > > > > > > ### Author Response · Authors · 2022-08-08
> > > > > > > **Thanks for your reconsideration**
> > > > > > >
> > > > > > > Dear reviewer kjHC,
> > > > > > >
> > > > > > > Many thanks for responding to our response in time.
> > > > > > >
> > > > > > > As your original evaluation was rejection (score 3) and raised a big concern on the mismatch between our experiments and our theory, we are wondering if this is still the main reason to keep score 3 after reading our response. If not, we sincerely hope the reviewer may post the current main concern after reading our response. We guess that the reviewer may not need our further response since the reviewer did not post further questions so we will not respond any further. However it might be important for us to understand your thought to think about how to further improve our work.
> > > > > > >
> > > > > > > Many thanks,
> > > > > > > the authors

---

> > > > > > > ### Author Response · Authors · 2022-08-08
> > > > > > > **A short follow-up response**
> > > > > > >
> > > > > > > We just saw the reviewer-raised example "a probably stronger baseline for proxy-CO might be making SA/GA (or other solvers) differentiable by the black-box trick (Pogancic et al. ICLR 2020), and then training the proxy model end-to-end". Our previous response has made a detailed and rigorous response on why this method cannot be applied to our experiments.
> > > > > > >
> > > > > > > Here, we just give a quick response to this argument to clarify why the black-box trick is an irrelevant baseline. The black-box paper is to learn a mapping from the data features to the parameters of a CO problem, while the solver to the CO problem, as the reviewer also mentioned, could be any solver but a given (no-need-to-learn) solver, i.e., the "black box" in the black-box trick.  And, it is not to learn a CO solver, the our focus of this work. Given this, we do not agree that the black-box paper can be a stronger baseline. One direct evidence for our argument is that the black-box trick cannot be applied to our experiments on circuit design, as in these applications, there are no downstream CO solvers to the original design problem as their CO objectives are entirely inaccessible. The methods SA/GA we used in the paper are to run on the proxy objective that has been learned.
> > > > > > >
> > > > > > > We sincerely hope the reviewer can check our response again and re-evaluate our work. We greatly appreciate for the effort!

---

> ### Comment · Reviewer_kjHC · 2022-08-09
> **Can you update the pdf?**
>
> Thanks to the authors for the discussions. I want to start a new discussion session so that we are not going down to too specific tiny points. As I write in my earliest review, the ideas/insights/experiments are novel and interesting, but this paper reads somewhat confusing. I can see new results provided in the rebuttal period, so can the authors update the pdf accordingly? **Talk is cheap**, since this year NeurIPS offers the pdf revision button, I will be happy to see the authors update the pdf, and I will re-evaluate this paper based on the revised version.
>
> Specifically, I will be happy to see the following issues addressed in the revised pdf:
> * Numerical results on pure CO problems compared with EGN. I read that EGN is a special case of your "principled design", so there should be other special cases that are worth discussion and comparison (seems that your response to Reviewer BQ6N can be included).
> * The "unsupervised" vs "supervised" issue. Personally, it makes me confused when reading this paper because proxy-CO is supervised. It makes more sense to drop the "unsupervised" term.
> * Clearer comparison with EGN (perhaps in a way similar to Table 1, if the current Table 1 could not distinguish them).

---

> > ### Author Response · Authors · 2022-08-09
> > **Thank you for the comments**
> >
> > Thank you so much for your response.
> >
> > We have started and are currently working on the paper revision. Note that we have already obtained and added the additional experiment results in the responses to the reviewers. We sincerely agree that these points above (the numerical results in pure CO problems, the supervised or unsupervised confusion, to highlight the relation to the baseline [1] in pure CO problems that naturally follow our principle) which might cause confusions should be further addressed and emphasized in the paper, but we feel really very sorry that we might not be able to finish this due to the time limitation and due to the page limit (the current space is kind of intense and we really need the additional page to emphasize these points). If we have any chance later than this period, we could further upload the revised version.
> >
> > Since we now already have the numerical results of the supplementary experiments during the rebuttal, and we now clearly know the three specific points that the reviewer mentioned above to clarify and address in the paper, adding these results and revising might not be that technically difficult, thus we will definitely address these issues as clear as possible, and enclose all of the results to the appendix of our final version.
> >
> > Thank you again for your response and comprehension.

---

> > > ### Comment · Reviewer_kjHC · 2022-08-10
> > > **Score raised**
> > >
> > > I appreciate the new results on pure CO, and I understand that the authors may not provide a revision due to limited time. Since there is no revised pdf, I am unsure some writing issues (such as "unsupervised" vs "supervised") will be adequately addressed in the final version. I raise the score by 1 in consideration of both.

---

### Official Review · Reviewer_BQ6N · 2022-07-10

**Rating:** 7
**Confidence:** 3
**Soundness:** 3 good
**Presentation:** 3 good
**Contribution:** 3 good

**Summary:**

The paper proposes an extension and variant to a previous paper by Karalias& Loukas, replacing the expectation based decoding strategy and probabilistic guarantee with another construction which relies on elementwise concavity+rounding of the function used to minimised the selfsupervised objective. The paper compares agains naive rounding, RL and Gumble softmax + sampling/rounding baselines on edge covering, node matching and 2 additional tasks (resource allocation and approximate programming) and finds their methods to be dominant across all considered methods.

**Questions:**

1. Most importantly: Is your method applicable to the problems studied in Karalias&Loukas? If so, can you add those experiments as well to directly compare the most relevant Literature method?
2. I saw you ran 3 seeds, can you add std to the table (and ideally, run additional seeds and perform a statistical significance test between methods?)

**Limitations:**

Authors discuss binary embeddings of the optimization variables as an avenue for future work. I would like to have seen a discussion of which functions might lie outside the entry-wise convex framework, and about the computational complexity and scaling of their proxies when moving to large scale problems

**Strengths And Weaknesses:**

The paper is well written and provides good experimental evidence of the improvement possible with their method, evaluated on real world problems. Originality, quality/clarity and significance are all at acceptance threshold for me.
The only nit picks I could find are in Questions and limitations.

---

> ### Author Response · Authors · 2022-08-02
> **Response to reviewer BQ6N**
>
> Thank you so much for reviewing our paper, appreciating our work and raising valuable comments. We address your questions as follows.
>
> > Comparing directly against [1] on its established benchmarks
>
> The reason we did not compare against our most relevant work [1] based on the benchmarks was explained in the above unified response. We add some comparison on pure (non-proxy-based) CO problems, edge covering and node matching, as shown above.
>
> > Significance of the results
>
> We would add the std value and related statistical analysis of all the experiments in the final version with the additional page. We are unable to do this during the rebuttal period due to the page limitation. In this response, we would like to to emphasize that according to our observation, there is no significant difference among the three random seeds in our cases (e.g. we introduce the results of the three different random seeds of our CON and AFF methods in application III in the following table).
>
> |            |         |   AFF   |         |         |   CON   |         |
> |:----------:|:-------:|:-------:|:-------:|:-------:|:-------:|:-------:|
> |  Threshold | seed #1 | seed #2 | seed #3 | seed #1 | seed #2 | seed #3 |
> | 3 AC units |  3.098  |  3.117  |  3.091  |  3.180  |  3.219  |  3.154  |
> | 5 AC units |  5.423  |  5.358  |  5.377  |  5.168  |  5.109  |  5.132  |
> | 8 AC units |  10.064 |  10.035 |  10.022 |  10.185 |  10.150 |  10.178 |
>
> > limitations of the current method
>
> The limitation questions asked by the reviewer are to the point. We think that every discrete function with binary inputs can be relaxed as an entry-wise concave function (if no constraint on the used order of moments), because of Thm. 2. However, if we only allow using lower-order moments, there might be a model capacity issue. So, we believe understanding the capacity and complexity of a lower-order entry-wise concave proxy model to learn an objective with higher-order moments is an important problem in the future.
>
> We also believe evaluating over large-scale problems is crucial. We are working towards this direction. We would like to highlight that the unsupervised learning framework shows great potential to scale up to large-scale problems. We have already observed some meaningful results (far better than Gurobi) on the max clique problem over a graph with 1k+ nodes.
>
> [1]Karalias et al. Erdos goes neural: an unsupervised learning framework for combinatorial optimization on graphs. NeurIPS 2020

---

> > ### Comment · Reviewer_BQ6N · 2022-08-07
> > **Thanks for the answers**
> >
> > Thanks for addressing my questions.
> >
> > In regards to questions raised by the other reviewers, and a response by you stating
> >
> > >We would like to highlight that the unsupervised learning framework shows great potential to scale up to large-scale problems. We have already observed some meaningful results (far better than Gurobi) on the max clique problem over a graph with 1k+ nodes.
> >
> > I would like to ask a clarifying yes or no question: If applied to the Erdos-goes-neural benchmark problems, does your algorithm recover their performance?  It sounds like you are able to evaluate it on it, and if your method is the general principle it feels like this is still the most relevant baseline (apologies if I missed it in the updated materials, but I have not noticed it anywhere)

---

> > > ### Author Response · Authors · 2022-08-08
> > > **Further response to reviewer BQ6N**
> > >
> > > The simple answer is yes.
> > >
> > > More extensive answer is as follows. As we claimed, our method and [1] will share the same pipeline if **the relaxed objectives of pure CO problems satisfy our proposed principle.** As [1] indeed uses some objectives that satisfy our principle in the two benchmarks it studied, our method will reduce to [1]’s performance on these benchmarks. In the final version, we will provide the experiments on more non-proxy-based CO problems, such as those studied in the unified response, to show the importance of our principle and to further highlight the relation to the baseline [1] (We have not updated the current version as for the page limitation. We will add these comparisons when the extra page in the final version is allowed).
> > >
> > > [1]Karalia et al. Erdos goes neural: an unsupervised learning framework for combinatorial optimization on graphs. NeurIPS 2020

---

> > > > ### Comment · Reviewer_BQ6N · 2022-08-08
> > > > **Please share whatever numbers you have already**
> > > >
> > > > Thank you for the clarification, especially the yes and no answer is much appreciated.
> > > >
> > > > The second part of my question (based on the quoted response) was asking about the direct comparison to EGN you already seem to be able to do on max clique?  Frankly I find it a bit weird that you do not make that comparison (apologies if I have missed it, but please point me to it otherwise) and while I like the paper otherwise, this making me reconsider my score

---

> > > > > ### Author Response · Authors · 2022-08-09
> > > > > **Further response to reviewer BQ6N**
> > > > >
> > > > > Sorry for the caused confusion. The reason why we did not compare directly against [1] in pure CO problems on its benchmarks is that, in these two problems (max-clique and graph-partitioning), both the losses of these two problems satisfy our entry-wise concave principle, so **by default we assume that our method will give the same loss and pipeline as [1]’s method on these two problems**.
> > > > >
> > > > > And as a matter of fact, on the max-clique problem, we use the same loss and the pipeline in [1] on hard datasets (generated by RB model, following [2]) with 1k+ nodes and obtain the result in the following table (this means that [1] also achieves the same performance). In additional to the max-clique problem, we also design another loss function that follows our entry-wise concave principle for the new vertex covering problem ([1] does not provide the loss for this problem and direct comparison is not available). Results on RB1000 hard dataset are shown as follows
> > > > > (The number denotes the node number, for max-clique problem the bigger the better, for vertex covering problem, the smaller the better):
> > > > >
> > > > > |            Problem           |    Max clique   | Vertex covering |
> > > > > |:----------------------------:|:---------------:|:---------------:|
> > > > > |              AFF (ours)             | 26.296 +/- 6.45 |  10.84 +/- 1.44 |
> > > > > | Gurobi (**same inference time**) | 25.125 +/- 6.64 |  11.4 +/- 1.754 |
> > > > >
> > > > > The reason why we did not put these results on large-scale datasets is that we had not done these large-scale experiments before submission. We will put these results in the final version.
> > > > >
> > > > > [1]Karalias et al. Erdos goes neural: an unsupervised learning framework for combinatorial optimization on graphs. NeurIPS 2020
> > > > >
> > > > > [2]Xu et al. Random constraint satisfaction: Easy generation of hard (satisfiable) instances. Artificial intelligence, 2007 - Elsevier

---

> > > > > > ### Comment · Reviewer_BQ6N · 2022-08-09
> > > > > > **Thanks for the results**
> > > > > >
> > > > > > Thank you for sharing these results. It is quite early here so I need to take another look later but additional questions:
> > > > > >
> > > > > >
> > > > > > 1. Are they comparable to the EGN setup?  I.e. they are the same dataset and can make comparable claims?
> > > > > > 2. Just to make sure I don't misunderstand things: your method would recover EGN results, but would it result in the same losses and schemes or would it result in a separate scheme like that used for experiments in the paper? If the former, I think adding the derivation on how to convert one into the other or highlighting it might resolve some of the reviewers concerns

---

> > > > > > > ### Author Response · Authors · 2022-08-09
> > > > > > > **Thank you for the response**
> > > > > > >
> > > > > > > Thank you for the response.
> > > > > > >
> > > > > > > > Are they comparable to the EGN setup? I.e. they are the same dataset and can make comparable claims?
> > > > > > >
> > > > > > > Yes, they are comparable with the EGN setup. In [1], they report the performance on two RB dataset with around 100 nodes. The small scale makes it difficult to compete against Gurobi, while we follow the very same RB model in [1] to generate even harder instances with 1k+ nodes, and extend the same loss as max-clique and our loss for vertex covering (new) on the large scale testing benchmarks. We find that in the larger and harder instances, the unsupervised learning pipeline would tend to outperform Gurobi given the same inference time.
> > > > > > >
> > > > > > > >your method would recover EGN results, but would it result in the same losses and schemes or would it result in a separate scheme like that used for experiments in the paper?
> > > > > > >
> > > > > > > The former (if applied on the max-clique or graph partitioning problem in pure CO, two cases studied by EGN).
> > > > > > >
> > > > > > > [1]Karalias et al. Erdos Goes Neural: an Unsupervised Learning Framework for Combinatorial Optimization on Graphs. NeurIPS 2020

---

> > > > > > > > ### Comment · Reviewer_BQ6N · 2022-08-09
> > > > > > > > **Comparing against EGN Table 2, can you provide the approximation ratio? and if possible in the limited time, perform a comparable experiment?**
> > > > > > > >
> > > > > > > > So if I understand correctly, the closest thing would be comparing your method to Table 2. of the EGN paper. Checking that table "at equal inference times" EGN-fast matches Gurobi at roughly 1.54 s/g with EGN-fast 0.708 vs Gurobi 0.69, and EGN-accurate matches Gurobi at roughly 6 s/g (a bit more for EGN) and EGN-A 0.735 vs Gurobi 0.790.
> > > > > > > >
> > > > > > > > Can you provide the Gurobi-run-to-optimality solution as well? That will allow us to do a *rough* apples to apples comparison. As it is, I can see EGN-A vs Gurobi at same inference would have a ratio of $0.735/0.790\approx 0.94$ , EGA-F vs Gurobi $0.708/0.697=1.01$ and your method comes in at $25.125/26.296 \approx 0.96$ with roughly a $0.014$ gap to EGN-A and $0.05$ to  EGN-F, higher being better. The inference time matters of course, but unless I have an error in my reasoning, this speaks favorably to your method already. If you can share any relevant context and numbers available to you already that would help get a clearer idea of the comparability.

---

> > > > > > > > > ### Author Response · Authors · 2022-08-09
> > > > > > > > > **Further response**
> > > > > > > > >
> > > > > > > > > It seems that the reviewer's computation has an error. If we understand the ask correctly, according to our above table, our method v.s Gurobi should be $26.296/25.125 \approx 1.05$ (instead of $25.125/26.296$ ), which is better than 1.01 and 0.94.
> > > > > > > > >
> > > > > > > > > Also, we have a question on the motivation of this comparison. As our method and EGN in principle will have the same result for this pure CO max-clique problem (we have explained the reasons), we would like to know the reason & motivation to numerically compare these two methods.

---

> > > > > > > > > > ### Comment · Reviewer_BQ6N · 2022-08-09
> > > > > > > > > > **Apologies for a typo**
> > > > > > > > > >
> > > > > > > > > > Apologies, I had mixed up the variables in my quick calculation. Yes, that makes it look even better, however knowing the optimal solution to directly compare optimality ratios and also having the running speeds and evaluation on *the same data* would still be appreciated.
> > > > > > > > > >
> > > > > > > > > > The motivation is that understanding can be flawed and argued about, but experimental numbers don't lie. Even if in theory the results will be the same, seeing it confirmed in practice helps make your point. "In theory" and "in practice" can always differ due to numerical issues, running time etc.  So seeing that you do indeed recover behaviour makes your point stronger, especially if you have the numbers already.

---

> > > > > > > > > > > ### Author Response · Authors · 2022-08-09
> > > > > > > > > > > **Further response and experiments**
> > > > > > > > > > >
> > > > > > > > > > > Yes. We can try to do these experiments. Currently, we are not sure how long it will take to get the optimal solution over a graph with 1k+ nodes but we are happy to check next and compute the approximation ratio. We will also do evaluation by using the exact same setting that EGN used.
> > > > > > > > > > >
> > > > > > > > > > > We see the explanation on the motivation. Thank you so much! This is a really nice point. We will definitely follow and keep it in our mind.

---

> > > > > > > > > > > ### Author Response · Authors · 2022-08-09
> > > > > > > > > > > **Numerical results updated**
> > > > > > > > > > >
> > > > > > > > > > > > Results on the same benchmarks.
> > > > > > > > > > >
> > > > > > > > > > > Attached are the results of the max-clique problem on Twitter dataset (same as Table 1 in [1]) and the RB test dataset (same as Table 2 in [1]). Same evaluation metrics are utilized as [1].
> > > > > > > > > > > |      Method      |         Twitter         |         RB test         |
> > > > > > > > > > > |:----------------:|:-----------------------:|:-----------------------:|
> > > > > > > > > > > |     Badloss+R    | 0.844 ± 0.159 (0.41s/g) | 0.704 ± 0.097 (0.44s/g) |
> > > > > > > > > > > |      GS-Tr+S     | 0.896 ± 0.125 (0.43s/g) | 0.742 ± 0.073 (0.47s/g) |
> > > > > > > > > > > |      GS-Tr+R     | 0.927 ± 0.091 (0.41s/g) | 0.787 ± 0.062 (0.44s/g) |
> > > > > > > > > > > | EGN(accurate)[1] | 0.942 ± 0.111 (0.42s/g) | 0.799 ± 0.067 (0.46s/g) |
> > > > > > > > > > > |        AFF       | 0.946 ± 0.084 (0.41s/g) | 0.805 ± 0.055 (0.44s/g) |
> > > > > > > > > > >
> > > > > > > > > > > > We are not able to give the optimal ratio on hard instances with 1k+ nodes currently.
> > > > > > > > > > >
> > > > > > > > > > > Sorry that we are not able to give the optimal ratio but instead give the average number of the selected nodes on the hard instances generated by the RB model with 1k+ nodes, because on these hard large-scale instances, solving for the optimal solution takes the Gurobi around 2000 seconds per instance. We have 200 instances in the testing set, due to the time limit, we are not able to generate all of the optimal solutions.
> > > > > > > > > > >
> > > > > > > > > > > [1]Karalias et al. Erdos Goes Neural: an Unsupervised Learning Framework for Combinatorial Optimization on Graphs. NeurIPS 2020

---

> > > > > > > > > > > > ### Comment · Reviewer_BQ6N · 2022-08-09
> > > > > > > > > > > > **Thank you for the direct comparison**
> > > > > > > > > > > >
> > > > > > > > > > > > Dear authors, thank you for these results. I understand the difficulty of doing these experiments in time. I have no further questions or concerns at this moment.

---

### Official Review · Reviewer_odo9 · 2022-07-10

**Rating:** 4
**Confidence:** 2
**Soundness:** 2 fair
**Presentation:** 2 fair
**Contribution:** 2 fair

**Summary:**

The authors propose an unsupervised learning framework for combinatorial optimization problems. They evaluate this framework by solving a synthetic graph optimization problem.

**Questions:**

It would be really helpful if the authors can clarify my questions above.

Further, I believe it would be productive for the rebuttal, if the authors can clarify the the following question first -

Given an instance of a minimum edge covering problem [1] G = (V,E) - how would you propose to solve this with your framework?

[1] https://en.wikipedia.org/wiki/Edge_cover

**Limitations:**

I find the setup of finding solutions in binary encoding space to be quite restrictive and don't immediately see how this would generalize to any combinatorial optimization problem in a feasible way - for eg. network flow algorithms in which one has to assign a fractional amount of flow to a network arc.



**Strengths And Weaknesses:**

Overall, I found the paper quite confusing to follow .

The authors claim that they have an unsupervised learning framework, yet they seem to be using supervised and labelled data to learn proxy objectives for both cost and constraints.

> This work provides the first unsupervised framework for PCO problems

The motivation behind unsupervised algorithms for PCO is unclear since you already need supervised data to train the proxy functions.

Keeping the supervised vs unsupervised nitpicking aside, I don't follow how either AFF or CON are actually solving the problem or how the guarantees about rounding provided by the authors are valid - if the function being optimized is a proxy of the true loss function - how can rounding this solution have any guarantee on feasibility - since the learned loss function has no bound on its error?

Taking a step back, I don't fully grasp how the authors go from an instance of an edge covering problem G = (V, E) to a valid solution X \in {0, 1}^|E| that solves the edge covering - my best guess is that the authors try to train two GNNs - the first GNN is trained in a supervised fashion to predict the cost function f (and perhaps the constraint function g), and the second GNN is used to learn a Bernoulli distribution from which the authors will sample lots of X, and the X that minimizes the learned/proxy cost function will be rounded out using the procedure they described into the final solution.

Section 4.1 - Each observed configuration is paired with a solution X  - what is the methodology for picking this X for the supervised training? Did you'll sample a solution at random and evaluate its cost and constrain function for the training data?

The notation was a bit confusing at times -
- line 216 the data configuration is denoted by C but the GNN encoding is also denoted by C in line 224.
- line 225 - 2nd order moments -  I am not familiar with this notation in this context of GNNs - did you mean the number of successive applications of the message passing layers.

I tried going through the supplementary material but it seems that some of the READMEs are incomplete.

What problem is being solved in section 4.3? Minimize the weighted edge covering? Why do you even have to use LCO for this problem? Don't you just have to solve MNIST on the nodes' images and then solve edge covering? Why do the edge weights have to be learned? I don't understand the motivation for picking this problem to evaluate the framework proposed in the paper.

Further, 4x4 graphs are extremely tiny. While the total search space of these graphs can be huge, it doesn't necessary imply that the family of instances in the data set has a very complex combinatorial structure or that learning this problem is hard.

If this synthetic example is just for the sake of demonstration, it would have been much easier to follow were it solving just the vanilla edge covering problem.

-----

My current recommendation is a borderline reject but I remain hopeful that I fundamentally don't understand something in the paper and that the authors will clarify it.

---

> ### Author Response · Authors · 2022-08-02
> **Response to reviewer odo9 (1/2)**
>
> Thank you so much for reviewing our paper, and providing very detailed comments and actionable suggestions. We feel sorry for caused confusions. Next, we will address such confusions.
>
> First, we would like to highlight that we also have studied PCO problems for circuit design in Applications II and III, which seem to have been missed by the reviewer. Second, we would like to clarify a few concepts in case that the reviewer feels confused about. Learning for CO means using machine learning models to generate solutions for CO. Even if the CO objective is well known, say a minimum edge covering problem, learning for CO can be still applied. Learning for Proxy-based CO (PCO) means that there is an extra challenge where the CO objective may not have closed form and needs to be learned first.
>
> ----------------------------------------------------------------------------------------------------------------------------
>
> > The quick question on how our method can solve the minimum edge covering problem
>
> This problem belongs to learning for (non-proxy-based) CO.  Our method has three steps.
>
> First, as the discrete cost and the discrete constraint are both known, our principle suggests we first write a relaxed cost and a relaxed constraint for this problem such that the relaxed functions are entry-wise concave. In the minimum edge covering problem, these two relaxation functions are not hard to write.  A relaxed objective is a simple linear sum $\sum_{e}w_e\bar{X}_e$, where $w_e$ denotes the edge weight. A relaxed constraint just follows our Eq.(7)
>
> $\sum_{v\in V} \prod_{e:v\in e} (1-\bar{X}_e)$.
>
>
>
> Second, our method will use a GNN that takes the graph with edge weights as input and generates the soft solution $\bar{X}_e$ as output. We optimize this GNN by minimizing the following loss over each training graph:
>
> $\sum_{e}w_{e}\bar{X_e}+ \beta \sum_{v\in V} \prod_{e:v\in e} (1-\bar{X}_{e}).$
>
> Third, over a testing graph, we use the learned GNN to generate the soft solution $\bar{X}_e$. We use the rounding method in Definition 1 to get the final discrete solution.
>
> We show some results of minimum edge covering to compare our model v.s. baselines for this problem in the above unified response.
>
> ----------------------------------------------------------------------------------------------------------------------------
> Next, we try our best to resolve other confusions that the reviewer raised.
>
> > Since we need to learn the proxy of the cost (or the constraint) by using supervised learning, why name our method as unsupervised learning?
>
> Sorry for the caused confusion. We answered this question in the unified response. Specifically, we say “unsupervised learning“ to refer to the procedure of learning to get solutions to CO problems, which follows the common terminology of the field of learning for CO. The way to learn proxy models is indeed based on supervised learning. We will make this point very clear in the final version.
>
> > How either AFF or CON are actually solving the problem or how the guarantees about rounding provided by the authors are valid?
>
> Our main theoretical contribution Thm. 1 is to show that if a discrete cost and a discrete constraint are relaxed into entry-wise concave continuous functions,  a low value of Eq.(3) guarantees a low-cost feasible solution. This is for general CO problems, no matter whether the objective is a learned proxy model or not. Technically, we have such a guarantee because entry-wise concave continuous functions make sure that the rounding procedure monotonically decreases the value of Eq.(3).
>
> Of course, for those proxy models, current Thm. 1 does not consider the potential learning error of the proxy models at the discrete points. The significance of Thm.1 for  proxy models is that even if proxy models have learning error, as long as they are entry-wise concave continuous functions  (e.g., AFF/CON), the rounding procedure based on them can still monotonically decrease the value of Eq.(3). If there is indeed some learning error of proxy model, the error term can be trivially merged into the bound in Thm. 1 because of the above monotonicity. Moreover, because of such monotonicity, if the value of Eq.(3) before the rounding is less than a threshold, the value after rounding is still less than a threshold. Thm. 1 proves that a value of Eq.(3) less than a certain threshold gives a solution that is guaranteed to be valid. Therefore, AFF/CON is crucial to guarantee valid solutions. In contrast, if proxy models do not satisfy entry-wise concavity, even if they can precisely learn the ground-truth cost and constraint at those discrete points, there is no performance guarantee.

---

> > ### Author Response · Authors · 2022-08-02
> > **Response to reviewer odo9 (2/2)**
> >
> > > The setting of the edge covering problem and the way to train proxy models by sampling $X$ for each configuration $C$
> >
> > Yes. The reviewer’s understanding is correct. For PCO problems, we have two GNNs: One GNN is used to learn the cost (or the constraint) given a configuration $C$ and an assignment $X$ on $C$; The other GNN is used to generate the assignment $X$ given a configuration $C$. The first GNN is learned based on a set of $C$’s where each $C$ is paired with a set of sampled $X$’s.
> >
> > For the edge covering problem in the paper, as the constraint can be derived as Eq.(7), only the cost function is to be learned.
> >
> > >Why do you even have to use LCO for the edge covering problem? Don't you just have to solve MNIST on the nodes' images and then solve edge covering? Why do the edge weights have to be learned?
> >
> > This is a semi-synthetic experiment to justify the effectiveness of our principle to learn a proxy model. Although the ground-truth cost is a simple sum of some node-feature-based edge weights, we do not assume that we know such a specific form in prior. We also do not assume we know any labels of the MNIST node features. So, the reviewer’s suggested pipeline is not applied. As we only know the objective value given an assignment $X$ but not the closed form, we use AFF/CON to model such an objective. This application has been used to study black-box optimization [1].
> >
> > > line 224: “These representations do not contain $X$ and are given by GNN encoding $C$”
> >
> > This statement should be revised as “These representations do not contain $X$ and are given by the GNN that encodes $C$”. Sorry for the confusion.
> >
> > > line 225: “2nd-order moments”
> >
> > Here, “2nd-order moments” are used to refer to the product term as $\bar{X}_v\bar{X}_u$ in Eq.(4). As discussed in Sec. 3.3, the general binary-input function with $n$ binary inputs may contain n-order moments (shown in line 175). However, it is impossible to build a proxy model that considers all such higher-order moments.
> >
> > > Limitation of the binary encoding space
> >
> > We agree with the potential limitation of binary encoding and thus propose to study the generalization of it in the future. However, for this initial work, we think starting from binary encoding is sufficient, which can already represent many CO problems. Note that the work [2] that is mostly relevant to us also has only studied the binary embedding case. [2] considers an even narrow range of problems where the binaries are defined on nodes, and no proxy-based CO problems are considered. Moreover, our binary encoding setting has already been applied to several very important applications including some circuit design problems as shown in this work.
> >
> > [1]Pogancic et al. Differentiation of Blackbox Combinatorial Solvers. ICLR 2020
> >
> > [2]Karalias et al. Erdos goes neural: an unsupervised learning framework for combinatorial optimization on graphs. NeurIPS 2020

---

> ### Author Response · Authors · 2022-08-08
> **Further response to reviewer odo9**
>
> Thanks again for reviewing our paper and providing the detailed comments and actionable suggestions. We have made the corresponding responses in detail. We are enthusiastic about whether the reviewer has any further comments and whether our responses have addressed the questions raised by the reviewer.

---

### Official Review · Reviewer_4diE · 2022-07-11

**Rating:** 7
**Confidence:** 3
**Soundness:** 3 good
**Presentation:** 3 good
**Contribution:** 3 good

**Summary:**

The paper proposes an unsupervised machine learning method for combinatorial optimization (CO) problems. The focus is in particular on CO problems that have an expensive cost function, necessitating a learned surrogate model. The authors extend the work by Karalias & Loukas [1] to this so called Proxy-based CO setting, while still retaining theoretical guarantees that a feasible solution with low cost can be attained. This guarantee comes with certain assumptions about the ground-truth cost function (not too high order) and limitations on the neural network for the proxy. The method outperforms learning based baselines on three different non-standard benchmarks and appears to require less resources to train.

**Questions:**

It would be helpful to explicitly spell out the limitations of the current method. For example: What are the computation costs associated with n-th order proxy costs? Will it fail when the ground truth cost requires higher order moments? How well does this scale to larger problem instances (>1k nodes)?

For suggestions about the experiments, see above.

**Limitations:**

Potential societal impact is addressed in the appendix. Limitations should be expanded on, as for example suggested above.

**Strengths And Weaknesses:**

Strength:
- Unsupervised machine learning for CO (especially PCO) is an interesting and important research area. In particular, this work is a significant extension of [1].
- The paper is well written, and the main ideas are clearly presented
- The proposed method appears sound: it is accompanied by theoretical guarantees on achieving feasible solutions with costs upper bounded by the training loss (theorem 1) and there exist relaxations fulfilling the required constraint for the proxy cost function (theorem 2)

Weaknesses:
- The experiments are extensive (multiple different baseline models and different experiments), but do not compare directly against prior work on any established benchmarks. This makes it difficult to place the work in the current literature and gauge how well it does in comparison.
- The non-learned baselines (simulated annealing and genetic algorithm) were only applied to the 2nd experiment "Resource Allocation in Circuit Design", but in light of recently pointed out shortcomings of ML based approaches compared to non-learned methods, it would be interesting to see a more extensive comparison against these non-learned baselines. This also includes applying SA/GA for more iterations, to get better results and a better sense of the complexity of the investigated problems.

---

> ### Author Response · Authors · 2022-08-02
> **Response to reviewer 4diE**
>
> Thank you so much for reviewing our paper, appreciating our work and raising valuable comments. We address your questions as follows.
>
> > Comparing directly against prior work on any established benchmarks
>
> The reason we did not compare against the most relevant work [1] based on the benchmarks used in [1] was explained in the above unified response. We added some comparison on pure (non-proxy-based) CO problems, edge covering and node matching, as shown above.
>
> Our work makes more significant contributions on learning for PCO problems. To verify this statement, we have compared directly against the prior work [2] on the circuit design application using the benchmark proposed in [2]. To the best of our knowledge, there have been no systematic PCO benchmarks. Please remind us if the reviewer knows other PCO benchmarks. We greatly appreciate it and we are happy to compare over these benchmarks  in the final version.
>
> > ML based approaches v.s. non-learned methods
>
> We are not sure which reference the reviewer refers to when claiming the shortcomings of ML based approaches compared to non-learned methods. To the best of our knowledge, our understanding of this statement comes from [3], which focuses on the TSP problem.
>
> We do not think the above statement is still suitable in our case. Because for the TSP problem considered in [3],  ML-based approaches get defeated by non-learnable methods such as the ‘Concorde TSP solver[4]’ since the latter ones have been improved for years and have leveraged a lot of heuristic insights. However, for proxy-based CO problems, our primary focus, there are few heuristics that can be directly leveraged due to  inaccessible objectives. One can only use the most generic CO solvers, such as SA/GA, while these generic CO solvers may hardly beat ML-based approaches.
>
> The current iteration number of SA/GA is controlled to use the similar inference time as ML-based approaches for fair comparison. We are happy to incorporate more-iteration evaluations on SA/GA in the final version. We are unable to do that during the rebuttal period because to get the ground-truth evaluation of the circuit application takes a lot of time.
>
> > limitations of the current method
>
> The limitation questions asked by Reviewer 4diE are very to the point. We believe understanding the capacity and complexity of a lower-order entry-wise concave model to learn a high-order proxy is an important problem. We believe evaluating over large-scale problems is also crucial. We are working towards these directions.
>
> We would like to highlight that the unsupervised learning framework shows great potential to scale up to large-scale problems. We have already observed some meaningful results (far better than Gurobi) on the max clique problem over a graph with 1k+ nodes.
>
> [1]Karalia et al. Erdos goes neural: an unsupervised learning framework for combinatorial optimization on graphs. NeurIPS 2020
>
> [2]Wu et al. IronMan-Pro: Multi-objective Design Space Exploration in HLS via Reinforcement Learning and Graph Neural Network based Modeling.
> IEEE Transactions on Computer-Aided Design of Integrated Circuits and Systems
>
> [3]https://openreview.net/forum?id=ar92oEosBIg
>
> [4]https://www.math.uwaterloo.ca/tsp/concorde.html

---

> > ### Comment · Reviewer_4diE · 2022-08-08
> > **Thanks for the response**
> >
> > Thank you for addressing my questions.
> >
> > > The current iteration number of SA/GA is controlled to use the similar inference time as ML-based approaches for fair comparison.
> >
> > The number of iterations applied to these two algorithms appears to be exceptionally low (e.g., 20 generations for GA). The fact that SA/GA do not even outperform the random baseline called HLS highlights this. Furthermore, it is unclear to me why the population size of GA is set to only 40, when the mini-batch size used during training of the proxies is 256. A simple and efficient baseline would simply evaluate as many random instances with the proxy as the GPU can fit into memory and pick the best (according to the proxy). This is akin to the HLS, which is only using 200 samples, though?
> >
> > Given the concerns by the other reviewers which also focus on the shortcomings of the experimental evaluation, albeit focusing more on the lack of comparison to EGN on pure CO problems, I believe more convincing baselines on demonstrably hard problems will greatly benefit the paper.

---

> > > ### Author Response · Authors · 2022-08-09
> > > **Further response to reviewer 4diE**
> > >
> > > Thank you for your further comments.
> > >
> > > First of all, we want to emphasize that **the current baselines GA/SA are also based on and guided by the trained proxies, instead of being guided by the HLS tools** which could give the real usage amount. Because directly utilizing HLS simulation tools for GA/SA evaluation is impossible from the aspect of time cost, inferring a single instance with a single set of assignments via HLS tools for the real usage amount takes almost fifteen minutes.
> > >
> > > Then, we further explain how we chose the hyper-parameters of the GA baseline. In application II, training the proxy and training the LCO optimization algorithm for convergence each takes around 120 seconds. In the testing period, using the GA (20 iterations with 40 populations) guided by the proxy without limits takes 48.774s per instance to evaluate on average, while using our framework guided by the CON proxy to infer takes 34.384s per instance on average. As training the LCO optimization algorithm needs some extra training time 120s, we allow GA using some more inference time 48s > 34s to make the overall time consumption of different methods comparable. The choice of GA using 20 iterations with 40 populations per instance was not to match the procedure of the training of the proxy model but to match the per-instance inference of our LCO algorithm $\mathcal{A}_{\theta}(C)$. Also, neither the GA method nor our LCO algorithm does the inference in a mini-batch fashion.
> > >
> > > But we think the reviewer’s suggestions are very to the point, we indeed have not noticed that GA can run multiple populations in parallel on GPU. Thus, we are now running further experiments on larger iterations and populations for the GA method. And we would give the performance-inference time relation table a little bit later. (Evaluating the real LUT, DSP usage via HLS tools takes too much time, and thus we will use our CON proxy to guide both our framework and the GA baseline, and report the predicted LUT/DSP usage of the GA method and our framework by the same CON proxy for fairness).

---

> > > ### Author Response · Authors · 2022-08-09
> > > **Results updated**
> > >
> > > We sincerely agree with the reviewer’s suggestions on the setting of the GA/SA baselines. Our current implementation of GA/SA is in a sequential manner during the proxy inference. We agree that for fairness the inference by proxy of each generation of the GA baseline should be conducted in parallel, which should be align with the LCO learning strategy. We are now working on the modification of our implementation to support GPU parallel in the GA baseline. We feel very sorry if such results could not be updated before the discussion deadline.
> > >
> > > But we have finished the experiments in sequential manner with much larger iterations. The LUT usage amount predicted by the CON proxy under 45%, 55%, 65% DSP maximum usage amount limits are shown below. Though the time of the GA with 600 iterations is extremely high (1672 seconds per instance in the sequential manner), it seems to potentially share the same time budget with CON if the GA baseline with these hyper-parameters were conducted under the GPU parallel implementation, yet the performance of GA in 600 iterations still has a gap from our method.
> > >
> > > | Method | Iteration | Population | Time(s) / instance | 45% DSP | 55% DSP | 65% DSP |
> > > |:------:|:---------:|:----------:|:---------------:|:-------:|:-------:|:-------:|
> > > |   GA   |     20    |     40     |        48       | 2493.84 | 2236.43 | 1967.53 |
> > > |   GA   |    600    |     40     |        1672         |    2079.94     |    1814.48     |    1619.11     |
> > > |   CON  |     -     |      -     |        34       | 1433.64 | 1116.17 | 1033.03 |

---

> > ### Comment · Reviewer_4diE · 2022-08-09
> > **Unsupervised vs supervised**
> >
> > The reviewers odo9 and kjHC hint at an interesting point when bringing up the additional supervision that is required in comparison to previous unsupervised CO methods: What is the distribution of the training samples used by the proxy NN? Does the optimizer find better solutions than are available in the training set? If so, how much better?
> >
> > Offering more insight into the training data required for the proxy models appears to be a good way to address the concerns around the additional supervision that is necessary in this work.

---

> > > ### Author Response · Authors · 2022-08-09
> > > **Response to Reviewer 4diE's comments on "unsupervised learning v.s. supervised learning"**
> > >
> > > We greatly thank reviewer 4diE for further commenting on this point.
> > >
> > > First. in case that it causes confusion, we would like to clarify that **it is not that our work brings up additional supervision compared to previous unsupervised CO methods.**   For learning for proxy CO, there have been no unsupervised CO methods to the best of our knowledge. Supervision to learn CO objectives is always needed for proxy CO problems regardless of the methods. The relevant work [1] is for unsupervised learning for non-proxy CO. For non-proxy CO, our framework also does not need any supervision to learn CO objectives. So, the supervision to learn a proxy model is not an extra requirement of our method compared to previous methods, but due to the CO problem itself.
> > >
> > > Regarding the questions, we have responses as follows.
> > > >"What is the distribution of the training samples used by the proxy NN?"
> > >
> > > In all applications of this work, for each configuration $C$, we adopt uniformly random assignments $X$ in the discrete space {0,1}$^n$. So, there is no extra/biased information during this procedure. But we agree with the reviewer that perhaps injecting effective information in the training dataset is helpful to learn a better proxy model, while it is beyond the scope of the current work.
> > >
> > > > "Does the optimizer find better solutions than are available in the training set? If so, how much better?"
> > >
> > > Yes, we indeed observe better solutions than the best solution in the training set. For example, in application II on circuit design, in Fig. 4, the yellow-circle points (HLS) correspond to the best solutions on one testing configuration among the same number of uniformly random assignments (the same number as the training data to learn the proxy model). We may observe that applying any CO solvers can at least slightly outperform such solutions, and the gain of our method is the most significant.
> > >
> > >
> > >
> > > [1]Karalia et al. Erdos goes neural: an unsupervised learning framework for combinatorial optimization on graphs. NeurIPS 2020

---

> ### Author Response · Authors · 2022-08-08
> **Further response to reviewer 4diE**
>
> Thank you again for reviewing our paper, appreciating our work and raising the insightful comments. We have made detailed responses to the comments, and we would appreciate to know whether the reviewer has any further comments.

---

### Author Response · Authors · 2022-08-02
**Unified response to all reviewers (1/2)**

We thank the reviewers for the time to review our manuscript and the actionable and constructive suggestions.  Three reviewers (**Reviewers 4diE, BQ6N, kjHC**) appreciate the novelty and the theoretical analysis of this work. In particular, two reviewers (**Reviewers 4diE, BQ6N**) also have clearly captured our main idea and thus strongly support our work. Another two reviewers (**Reviewers odo9, kjHC**) feel confused about some statements of this work and raised insightful questions. We are sorry about the confusion and promise to address it in the final version (given the one additional page). Here, we first address two main questions in this unified response. We will further address other questions in the specific response to each reviewer.

>Why not compare with the prior method in [1] and use the established benchmarks in [1] on some pure (non-proxy) CO problems, since we have claimed improvement over [1]?

This confusion essentially comes from the ambiguity on *which aspects our method improves [1]*.

Our method does not improve [1] on *solving a pure CO if a relaxed objective that follows our proposed principle is given*. Actually, for a pure CO with a relaxed objective following our proposed principle, our method and [1] have the same pipeline, i.e., sequentially rounding the soft solutions. For the max clique and normalized cut problems, the two pure CO benchmarks studied in [1], the Erdos-expectation-based objectives used in [1] happen to have closed forms that are entry-wise affine and thus satisfy our proposed principle. So, it is meaningless to compare with [1] on these two pure CO benchmarks.

*Instead, our method improves over [1] in another three significant aspects, the proposal of relaxation principle, theoretical guarantees based on relaxation, and a generalization to PCO problems*. Specifically,

1) Our work is the first to propose the principle on how to relax a general CO objective (including pure CO or proxy-based CO) so that we do not need to use Erdos-expectation-based objectives (Eq. (2) in our paper or Eq. (3) in [1] as suggested in [1]) that are hard to compute for general CO problems. Note that although the ultimately-used objectives in the two cases in [1] happen to satisfy our principle, the work in [1] neither gave a principle as ours nor introduced how to derive an objective for general CO problems to satisfy such a principle. [1] adopted a case-by-case analysis, which cannot provide the generalizable knowledge given by this work.

2) In theory, we proved that even if we do not use the Erdos-expectation-based objective (Eq. (2) in our paper or Eq. (3) in [1]) while using a deterministic relaxation-based objective instead, we may still achieve a performance guarantee similar to that proposed in [1]. Such a deterministic relaxation-based objective is more friendly to compute for general CO problems (as we explained in Sec. 3.1). And, the theoretical bound is not probabilistic which means sampling is not needed to achieve the guarantee.

3) Because our principle is suitable for general CO problems, we can easily work on those more challenging proxy-based CO problems, where the objectives do not have closed forms and have to be learned.  We just need to set the proxy models to satisfy our principle. Because [1] did not provide the principle, [1]’s approach is unable to be applied to PCO problems.

[1]Karalias et al. Erdos Goes Neural: an Unsupervised Learning Framework for Combinatorial Optimization on Graphs. NeurIPS 2020

---

> ### Author Response · Authors · 2022-08-02
> **Unified response to all reviewers (2/2)**
>
> In the next bullet, we run an experiment on pure CO problems to compare different methods.
>
> >Comparison on pure CO problems
>
> As asked by **Reviewers 4diE, odo9, BQ6N, kjHC**, we further study the non-proxy-based edge covering and node matching problems. In this setting, we use the same 4 $\times$ 4 grid to represent the graph as those in application I in the paper, but we utilize the digit values (instead of images) directly as node attributes $C_u, u \in V$. We still set the cost as the sum of edge weights $f(X;C) = \sum_{e \in E} W_e X_e$, where $W_e=C_uC_v$ for $e=(u,v)$ but here we consider a pure CO problem, i.e., the cost is no longer needed to be learned.
>
> To emphasize the importance of our principle, we add another baseline called ‘Badloss+R’. In ‘Badloss+R’, the relaxed cost ($f_r’$) and the relaxed constraint ($g_r’$) do not satisfy our principle. Instead, this baseline imposes trigonometric functions to the original entry-wise concave functions $f_r’(\bar{X};C) = f_r(\sin(9\pi \bar{X}/2);C), g_r’(\bar{X};C) = g_r(\sin(9\pi \bar{X}/2);C)$, where those $\sin$ functions operate on each entry of the input vector. Note that in this case, these functions still satisfy $f_r’(X;C) = f_r(X;C), g_r’(X;C) = g_r(X;C)$ on the discrete points $X \in$ {0,1}$^n$, while $f_r’, g_r’$ oscillate a lot in the relaxation regime $\bar{X}\in (0,1)^n$.
>
> The results are shown as follows:
>
> |   Method  |   Edge covering  |  Node matching  |
> |:---------:|:---------------:|:---------------:|
> | Badloss+R | 272.52 $\pm$ 2.25 | 251.44 $\pm$ 2.74 |
> |     RL    | 189.48 $\pm$ 1.25 | 207.13 $\pm$ 1.04 |
> |  GS-Tr+S  | 210.91 $\pm$ 1.50 |        -        |
> |  GS-Tr+R  | 167.85 $\pm$ 0.74 | 201.52 $\pm$ 1.14 |
> | AFF(ours) | 166.28 $\pm$ 0.70 | 198.31 $\pm$ 0.89 |
> |  OPT(gt)  |    157.61    |    190.51   |
>
> The gaps among different methods in this pure CO case are roughly the same as the results of the corresponding PCO problems (application I) in the paper. Our method AFF still outperforms all the baselines. The poor performance of ‘Badloss+R’ reveals the importance of our principle to relax a CO objective.
>
> Here, we also notice that the gap between ‘GS-Tr+R’ and ‘AFF’ becomes smaller than that using proxy models in application I of the paper, because in this case, loosely speaking, the Erdos-expectation-based objective that  ‘GS-Tr+R’ can directly optimize will reduce to our relaxed objective because of the entry-wise affine objective given to this problem. So, here ‘GS-Tr+R’ essentially also benefits from our relaxation principle.
>
> >Since we need to learn the proxy of the cost (or the constraint) by using supervised learning, why name our method as unsupervised learning?
>
> We say “unsupervised learning“ to refer to the procedure of learning for CO, which follows the common terminology in the field of learning for CO. The way to learn proxy models is indeed based on supervised learning. We will make this point very clear in the final version.
>
> [1]Karalias et al. Erdos Goes Neural: an Unsupervised Learning Framework for Combinatorial Optimization on Graphs. NeurIPS 2020

---

### Meta-Review · Area_Chair_Jaim · 2022-08-26

**Recommendation:** Accept
**Confidence:** Certain

**Metareview:**

This paper proposes a new framework to train machine learning models to solve combinatorial optimization problems. Its key idea is to develop a relax-and-round solution-generating approach that allows backpropagation and enjoys theoretical guarantees for the rounding procedure. This approach generalizes the previous work (Karalia et al. 2020) to proxy CO settings where the objective function is expensive (and replaced by a proxy function).

The reviewers are all on the positive side of this paper. While the experiments are somewhat toy-like, all the reviewers appreciated the key idea and the theoretical results of this paper. I tend to agree and recommend acceptance for this paper.

However, I would strongly encourage the authors to improve their presentation for their final manuscript. Upon reading, the paper caused confusion on (1) using the terminology "unsupervised" and (2) its relationship to the prior work (Karalia et al. 2020). Although this concern has been resolved after a thorough discussion between reviewers and authors, I would be happy to see the corrections being made for the final manuscript.

**Award:**

No

---

### Decision · Program_Chairs · 2022-09-14

Accept